# Factorized Scheduling Principle: Learning Interpretable and Transferable Policies via Structured Additive Functions

**Hong Je-Gal** [1]  **Hyun-Suk Lee** [1 2]

## Abstract

Scheduling problems arise from repeatedly selecting one item from a set of candidates based on their states. These problems often reduce to assigning priority scores and choosing the highest-ranked item. In this work, we propose a factorized scheduling principle (FSP) framework to learn interpretable and transferable scheduling rules. The FSP framework represents system states as condition distributions and decomposes a global scheduling principle into additive univariate and pairwise components with identifiability constraints. The scheduling principle enables the framework to maintain a simple priority-based structure during deployment. This principle is learned by using a policy-based objective combined with a temporal-difference signal defined on the condition distribution. Experiments on synthetic and realistic scheduling tasks demonstrate the FSP framework's strong performance, interpretability, and zero-shot generalization across different system scales.

## 1. Introduction

Scheduling is a fundamental decision-making problem that arises whenever a system must repeatedly select one item among a set of candidates. Regardless of the application domain, a large class of scheduling problems follows a common structure: at each decision point, the system evaluates the priority of candidate items based on their conditions and selects one with the highest priority (Shani et al., 2005; Lu & Yang, 2016; Ferrá et al., 2003; Wei et al., 2018; Stidham & Weber, 1993; Han & Liu, 2008). This structure suggests that scheduling could be reduced to simple priority-based rules

in principle (Haupt, 1989). In practice, however, deploying effective scheduling strategies remains difficult, particularly when system conditions or candidate populations vary.

Traditionally, classical index-based approaches, such as the Gittins and Whittle indices, formalize the concept of item-level priority rules and demonstrate that these rules can yield transparent and reusable decision strategies (Whittle, 1988; Liu & Zhao, 2010; Gittins et al., 2011). Their item-wise structure yields per-item priority scores, allowing the evaluation independent of the number or ordering of candidates. However, deriving such indices often relies on strong modeling assumptions and low-dimensional structures, which limits their applicability in modern high-dimensional environments (Scully & Terenin, 2025). Moreover, extending these indices to new systems usually requires manual redesign or heuristic adaptation (Maatouk et al., 2020; Robledo Relaño et al., 2024).

In contrast, recent learning-based approaches, particularly reinforcement learning (RL), provide greater flexibility by optimizing policies directly from experience data (Xu et al., 2017; Wei et al., 2018; Ye et al., 2019; Huang et al., 2021). These policies can achieve strong empirical performance in specific environments. However, they generally operate on joint state representations in which item features are encoded together, tying policy behavior to item ordering or a fixed candidate set size. This entanglement makes the learned decision rule difficult to interpret and brittle when the candidate set changes (e.g., when new items appear or the size varies). As a result, even if the underlying scheduling logic should conceptually remain similar, policies trained in one environment often fail to transfer to systems with different item populations (Lee et al., 2019).

These limitations reveal a gap between the classical and learning-based approaches. Classical approaches offer transparency and transferability but rely on restrictive assumptions, while learning-based approaches provide flexibility at the expense of interpretability and transferability. This raises an attractive but unexplored question: *Can scheduling principles themselves be learned from experience data, while remaining interpretable and reusable across systems?*

In this work, we propose a factorized scheduling principle (FSP) framework to address this question for item-selection

[1]Department of Artificial Intelligence and Robotics, Sejong University, South Korea [2]Artificial Intelligence Robotics Institute (AIRI), Sejong University, South Korea. Correspondence to: Hyun-Suk Lee <hyunsuk@sejong.ac.kr>.

*Proceedings of the 43rd International Conference on Machine Learning*, Seoul, South Korea. PMLR 306, 2026. Copyright 2026 by the author(s).

scheduling problems, where decisions are naturally governed by relative priorities among candidate items. The FSP framework learns a *global scheduling principle* defined over item-level features, rather than a black-box policy tied to a specific state representation or candidate set. Since the principle is defined on item-level features rather than joint state vectors, it naturally generalizes to different candidate sets without depending on item indices or state concatenation. The key idea is that scheduling decisions can be governed by how each item's conditions contribute to its priority, regardless of the identities or number of other items. The contributions of this paper are summarized as follows:

- We introduce a novel formulation that learns an interpretable and transferable scheduling principle directly from experience using complementary local and temporal signals.
- We design a structured additive representation that decomposes the scheduling principle into interpretable feature-level contributions and interactions, while ensuring identifiability through centering constraints.
- We provide theoretical analysis establishing identifiability, approximation guarantees, ranking consistency, and transfer properties of the learned principle.
- Through synthetic and realistic scheduling experiments, we demonstrate that the FSP framework achieves strong performance, interpretability, and zero-shot generalization across candidate populations.

**Conflict of Interest Disclosure.** The authors declare no financial conflicts of interest related to this work.

## 2. Background and Motivation

### 2.1. Motivation for Learning Scheduling Principles

A wide range of real-world systems rely on scheduling to select an item from a set of candidates, e.g., wireless base stations choosing which user to serve (Ferrá et al., 2003; Wei et al., 2018), operating systems deciding which process to execute, manufacturing lines prioritizing jobs (Ouelhadj & Petrovic, 2009), logistics platforms dispatching tasks to workers (Delaram & Valilai, 2018; Su & Dong, 2025), and recommender systems selecting an item to display (Shani et al., 2005; Lu & Yang, 2016; Huang et al., 2021). Traditionally, scheduling has relied on hand-crafted heuristics (e.g., shortest-job-first in computing and proportional fairness in wireless networks) or analytically derived priority rules (e.g., Gittins index and Whittle index). They are effective and transparent in their intended settings (Liu & Zhao, 2010; Maatouk et al., 2020), but these heuristics are inflexible; their logic does not easily transfer across systems and often requires careful re-engineering. RL is a flexible alternative that optimizes policies directly from experience data, achieving strong empirical results across application domains (Xu et al., 2017; Wei et al., 2018; Ye et al., 2019). However, such RL-based schedulers are often make it difficult to interpret their decision logic and sensitive to changes in system configurations or candidate populations (Lee et al., 2019; Milani et al., 2024).

These limitations highlight the need for a different paradigm: learning explicit scheduling principles that operate directly on item-level features and remain valid across systems within the same application domain, rather than relying on entangled black-box policies tied to a specific state or candidate set. Such a scheduling principle should make transparent how each feature and feature interaction contributes to an item's scheduling priority, thereby inducing a consistent ordering over items regardless of the size or composition of the candidate set. The FSP framework introduced in this paper is designed to achieve this goal. It represents scheduling priorities as a structured additive function over descriptive features. The FSP framework combines the adaptability of learning-based methods with the interpretability and transferability of classical scheduling heuristics.

### 2.2. Standard Scheduling Problem

We model a standard scheduling problem as a discrete-time Markov decision process (MDP) in which, at each step, a system selects an item from the candidate set. Specifically, at time step $t$, the environment presents a set of candidate items $\mathcal{N} = \{1, \cdots, N\}$. Each candidate item $n$ has a feature vector $\mathbf{x}_n^t = (x_{n,1}^t, \cdots, x_{n,K}^t) \in \mathcal{X}$ (e.g., channel quality, demand level, and queue length), where $\mathcal{X} = [0, 1]^K$ is a feature space[1]. The system state is defined as $s^t \in \mathcal{S}$, which consists of all the information required for scheduling. Typically, it is defined as a concatenated vector of the candidate items' features $(\mathbf{x}_1^t, \cdots, \mathbf{x}_N^t) \in [0, 1]^{K \times N}$, which implicitly ties the state representation to both the number and the ordering of candidate items.

The action is defined as $a^t = n \in \mathcal{N} = \mathcal{A}$, where $n$ is the index of the chosen item. Scheduling the item $a^t$ yields a reward $r(s^t, a^t)$ and transitions to a next state distribution $s_{t+1} \sim P(\cdot|s^t, a^t)$. The goal of the problem is to find a policy $\pi(a|s)$ that maximizes the expected discounted return:

$$\max_\pi J(\pi) = \mathbb{E}_\pi \left[ \sum_{t=0}^{\infty} \gamma^t r(s^t, a^t) \right], \quad (1)$$

where $\gamma$ is a discount factor.

This standard scheduling formulation is convenient because each candidate item is directly represented by an index in the action space. For this reason, it is widely adopted across a broad range of application domains. (Stidham & Weber, 1993; Chang et al., 2000; Ferrá et al., 2003; Shani

---

[1]Without loss of generality, we assume that feature vectors are normalized to $[0, 1]^K$.

et al., 2005; Han & Liu, 2008; Lu & Yang, 2016; Xu et al., 2017; Wei et al., 2018; Ye et al., 2019; Huang et al., 2021). However, it suffers from several critical limitations:

- **No explicit principle**: The learning objective focuses on estimating full state-action value functions $Q(s, a)$ or policies $\pi(a|s)$ that depend on the entire candidate set and its joint configuration. As a result, the effect of an individual item's feature vector cannot be isolated into a shared, item-level priority rule. This prevents the emergence of a global scheduling principle that can be consistently evaluated across different candidate sets.
- **Lack of interpretability**: The learned $Q$ function or policy is a high-dimensional black-box mapping from a joint state representation to actions. Since the contribution of individual features or feature interactions is not explicitly parameterized, it is difficult to attribute scheduling decisions to specific feature conditions or to analyze the learned behavior analytically.
- **Poor transferability**: Since actions are defined by item indices and the policy is trained on a fixed candidate set size $N$, the learned strategy is inherently coupled to the training distribution of candidate sets. Even if the underlying task structure remains unchanged, the policy may not preserve the same scheduling strategy, if the number of items, their ordering, or their feature distribution changes.

These limitations motivate our alternative framework, which factorizes scheduling decisions over item features and their interactions. Rather than learning an index-specific action map tied to a fixed candidate set, FSP learns a priority rule defined in feature space, allowing it to be evaluated on arbitrary candidate sets and yielding an interpretable, transferable global scheduling principle.

## 3. Proposed Framework: Factorized Scheduling Principle

Here, we present an FSP framework built on the idea that scheduling policies should be guided by an interpretable and transferable *global scheduling principle* $S : \mathcal{X} \to \mathbb{R}$, rather than by opaque black-box models. Beyond the empirical evidence from classical heuristics, the structure of scheduling problems naturally suggests the existence of such a principle: decisions are made by selecting an item based on its feature conditions, which largely determine the resulting reward. Accordingly, the priority of an item can be described as a function $S(x)$ of its feature vector, which is evaluated consistently regardless of the particular candidate set.

In what follows, we describe the FSP framework by detailing (i) how the state is represented as a distribution of item features, (ii) how the global principle is defined as a structured additive function, (iii) how candidate items are scored

and selected, and (iv) how the global principle is learned from experience. We provide a high-level overview of the FSP framework in Figure 1. The following subsections introduce each component in detail.

### 3.1. Condition Distribution as State Representation

At time step $t$, the system is characterized by a collection of item-level feature vectors $\{\mathbf{x}_n^t\}_{n \in \mathcal{N}}$, where $\mathbf{x}_n^t \in [0, 1]^K$ describes the condition of item $n$. However, as mentioned in Section 2.2, this representation is not invariant to the identity or ordering of items. Therefore, a state representation is needed that can appropriately describe the scheduling priorities over items while being invariant to permutations and accommodating variable-sized candidate sets.

To this end, we express the state as a *condition distribution* over the feature space:

$$\bar{s}^t(\mathbf{x}) = \sum_{n=1}^{N} \delta(\mathbf{x} - \mathbf{x}_n^t),$$

where $\delta(\cdot)$ is a Dirac delta function. This captures the current population of items as point masses at their feature locations. Since scheduling priorities are functions of item feature conditions, it is natural to represent the system state in terms of the distribution of item features. Furthermore, unlike a concatenated feature vector $(\mathbf{x}_1^t, \cdots, \mathbf{x}_N^t)$, this distributional representation remains valid regardless of the number of items, $N$, and it abstracts away item identities, including their indices, while retaining the relevant conditions for scheduling. In practice, each delta function can be replaced with a smooth kernel bump function $\kappa(\mathbf{x}, \mathbf{x}_n^t)$, yielding

$$\bar{s}^t(\mathbf{x}) = \sum_{n=1}^{N} \kappa(\mathbf{x}, \mathbf{x}_n^t).$$

This smooth density field avoids discontinuities, supports differentiable approximations, and interacts well with basis-function parameterizations of the global principle. Consequently, this condition distribution $\bar{s}^t(\mathbf{x})$ can describe candidate items in a stable and transferable manner. It is not intended to redefine the underlying MDP, but rather to provide a transferable description of candidate conditions.

### 3.2. Global Scheduling Principle

We define a function $S : \mathcal{X} \to \mathbb{R}$, called the global scheduling principle, that encodes the priority associated with any feature condition $\mathbf{x}$. Unlike state-specific value functions from the standard MDP perspective, this principle is intended to capture general rules of scheduling that can be applied across different candidate sets and environments. The goal of the FSP framework is to learn $S(\mathbf{x})$ in a way that explains why certain items are preferred under specific feature conditions, and that generalizes across systems with

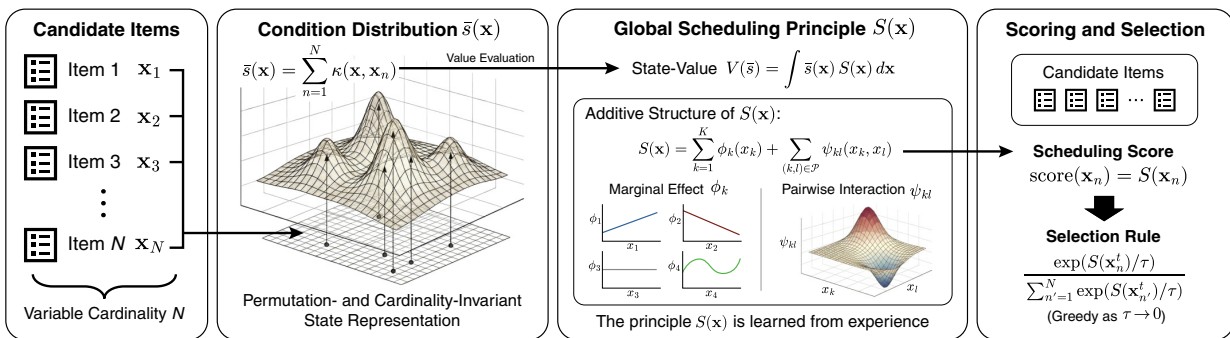

*Figure 1.* Illustrative overview of the FSP framework.

different numbers or identities of items. First, we list design goals that a good structure of $S$ should satisfy:

- **Continuity**: Representing smooth effects of feature levels, avoiding discretization artifacts.
- **Interpretability**: Decomposing into components that can be visualized as human-readable geometrization.
- **Transferability**: Remaining applicable when the number or identity of items changes.
- **Identifiability**: Ensuring that marginal and interaction effects are uniquely separated without overlap.

To achieve these goals, we introduce a structured additive formulation with identifiability constraints. The additive structure introduces an explicit inductive bias toward reusable, item-level scheduling rules, while remaining expressive enough to capture feature interactions. Its smooth one- and two-dimensional components provide continuity and interpretability, while its definition on item-level features ensures transferability. The centering constraints guarantee identifiability. We parameterize $S$ as a structured additive function as

$$S(\mathbf{x}) = \sum_{k=1}^{K} \phi_k(x_k) + \sum_{(k,l) \in \mathcal{P}} \psi_{kl}(x_k, x_l),$$

where $\phi_k$ are one-dimensional curves (marginal effects for each feature), $\mathcal{P}$ denotes the set of feature pairs with interactions, and $\psi_{kl}$ are two-dimensional surfaces (interactions for selected pairs). To ensure identifiability (i.e., uniquely separate marginal and interaction effects), we consider centering constraints under the measure on $[0, 1]$ as follows:

$$\text{(mean)} \quad \int_0^1 \phi_k(x_k)\, dx_k = 0, \tag{2}$$

$$\text{(row)} \quad \int_0^1 \psi_{kl}(x_k, x_l)\, dx_k = 0, \tag{3}$$

$$\text{(column)} \quad \int_0^1 \psi_{kl}(x_k, x_l)\, dx_l = 0, \tag{4}$$

$$\text{(global)} \quad \int_0^1 \int_0^1 \psi_{kl}(x_k, x_l)\, dx_k dx_l = 0. \tag{5}$$

These centering conditions ensure that $\psi_{kl}$ captures only

pure interactions beyond the one-dimensional marginals $\phi_k$, without ambiguity from additive shifts.

In practice, we can use smooth function bases, such as spline or lattice parameterizations, to parameterize the one- and two-dimensional components $\{\phi_k\}$ and $\{\psi_{kl}\}$. These parameterizations are well-suited for representing continuous feature effects, enable direct visualization of marginal and interaction components, and admit well-established approximation guarantees. It is worth noting that the choice of spline or lattice bases is not essential, but provides a concrete instantiation that supports both interpretability and the theoretical analysis presented later.

### 3.3. Scheduling Scoring and Selection Rule

Scheduling requires evaluating a variable set of candidate items in a consistent way; using the global principle $S(\mathbf{x})$ over the feature space, each item can be scored directly by its feature vector. For a candidate item $n$ with $\mathbf{x}_n^t$, the scheduling score is simply given by

$$\text{score}(\mathbf{x}_n^t) = S(\mathbf{x}_n^t).$$

This connects the global scheduling principle to the local decision: the policy prefers items whose feature conditions are globally assessed as high priority. Then, we can define the value of a condition distribution $\bar{s}^t$ as the sum of item-level scores:

$$V(\bar{s}^t) = \int \bar{s}^t(\mathbf{x})\, S(\mathbf{x})\, d\mathbf{x} = \sum_{n=1}^{N} S(\mathbf{x}_n^t),$$

where alternative normalizations (e.g., averaging) can be used without changing the induced item ranking. This ensures that the evaluation of states is consistent with the evaluation of their constituent items.

With this scheduling scoring, we adopt a softmax policy over item scores as

$$\pi(n|S, \{\mathbf{x}_n^t\}_{n \in \mathcal{N}}) = \frac{\exp(S(\mathbf{x}_n^t)/\tau)}{\sum_{n'=1}^{N} \exp(S(\mathbf{x}_{n'}^t)/\tau)}, \tag{6}$$

where $\tau$ is the temperature parameter that balances exploitation and exploration. As $\tau \to 0$, the policy becomes greedy

and selects the highest-scoring item, while larger values of $\tau$ induce stochasticity, allowing exploration of alternative actions. In this policy, every decision can be explained by the feature levels or interactions that contribute to the item's priority since each score decomposes via the functions $\phi_k$ and $\psi_{kl}$, making the policy transparent. We can inspect why a particular item was chosen by linking it directly to the interpretable components of the global principle.

### 3.4. Learning the Global Scheduling Principle

In the FSP framework, the goal of training is not to obtain an unbiased policy gradient estimator, but to directly shape the global scheduling principle $S(\mathbf{x})$ through complementary local and temporal signals. Instead of training a separate policy and value function, the framework derives them from the same principle, making the learning process more interpretable.

At each time step $t$, an experience is obtained, which is defined as a tuple $(\bar{s}^t, n^t, r^t, \bar{s}^{t+1})$ of the current condition distribution, chosen item, observed reward, and the next condition distribution. Using the experience, we employ two complementary loss terms to learn the global scheduling principle $S$. First, a policy likelihood loss encourages $S$ to assign high scores to actions that yielded high reward:

$$\mathcal{L}_{\text{lik}} = -r^t \log \pi(n|S, \{\mathbf{x}_n^t\}_{n \in \mathcal{N}}),$$

where $\pi$ is the softmax policy induced by $S(\mathbf{x}_n^t)$'s. Also, a value regression loss enforces temporal consistency of the global value $V(\bar{s}^t)$ with temporal difference targets:

$$\mathcal{L}_{\text{val}} = (y^t - V(\bar{s}^t))^2,$$

where $y^t = r^t + \gamma V(\bar{s}^{t+1})$ is a target value. Furthermore, to maintain interpretability and identifiability, we consider the structural regularization terms, including $l_2$ penalties on the coefficients of additive functions to control complexity and centering penalties of $\psi_{kl}$ to ensure identifiability.

The training loss for the global scheduling principle $S$ is given by

$$\mathcal{L} = \mathcal{L}_{\text{lik}} + \lambda \mathcal{L}_{\text{val}} + \Omega(\phi, \psi), \tag{7}$$

where $\lambda$ is a hyperparameter that controls the trade-off between the likelihood loss and the value regression loss and $\Omega$ collects regularization terms. The parameters of the structured additive function $S$, i.e., $\{\phi, \psi\}$, are updated via stochastic gradient descent using the loss. This procedure directly trains the global scheduling principle, yielding both interpretability and transferability.

**Why combine the two losses?**   We consider these two losses since they provide complementary learning signals. $\mathcal{L}_{\text{lik}}$ shapes the *local ranking of items* within each condition distribution by directly linking observed rewards to chosen

actions. On the other hand, $\mathcal{L}_{\text{val}}$ enforces *global consistency* across time by aligning state values with long-term returns. Using only policy likelihood loss captures immediate preferences across the candidate items, but it ignores future consequences. Using only value regression loss yields consistent state values, but it may not prioritize items correctly. By combining both losses, the learned principle $S(\mathbf{x})$ faithfully reflects both local action-level preferences and global temporal structure.

### 3.5. Theoretical Analysis

We provide theoretical analysis of the FSP framework with respect to interpretability, performance, and transferability. We assume the existence of the ground-truth principle $S^\star$ and establish guarantees under mild regularity assumptions. These assumptions are used to state clean theoretical guarantees. In realistic coupled systems, an exact additive ground-truth principle may not exist; in such cases, FSP should be interpreted as learning an approximate surrogate principle that captures the dominant reusable priority structure. The detailed technical assumptions and proofs are provided in Appendix B.

**Interpretability.**   A requirement for interpretable scheduling principles is that their decomposition into one- and two-dimensional components be well-defined and meaningfully inspected as shown in Theorem 3.1.

**Theorem 3.1** (Identifiability). *Under the centering constraints in Equations* (2)–(5), *the additive scheduling principle* $S(\mathbf{x}) = \sum_k \phi_k(x_k) + \sum_{(k,l) \in \mathcal{P}} \psi_{kl}(x_k, x_l)$ *is unique.*

**Performance.**   The approximation theorem in Theorem 3.2 shows that the concrete instantiations of the structured additive principle, such as spline or lattice parameterizations, can approximate any sufficiently smooth ground-truth principle with controllable error. Based on this precise approximation, the learned principle should preserve the ranking of candidate items and deliver near-optimal scheduling performance as stated in Theorem 3.3.

**Theorem 3.2** (Approximation error). *For any Lipschitz-continuous ground-truth principle* $S^\star$ *with bounded second mixed derivatives, there exists an additive scheduling principle* $\hat{S}$ *parameterized with spline and lattice basis functions such that* $\|S^\star - \hat{S}\|_\infty = O(R^{-2})$, *where* $R$ *is the resolution of the basis functions.*

**Theorem 3.3** (Ranking consistency & Greedy regret). *Suppose the learned principle* $\hat{S}$ *satisfies* $\|S^\star - \hat{S}\|_\infty \leq \epsilon$. *Then, for any features of the candidate items, if* $S^\star(\mathbf{x}_i) > S^\star(\mathbf{x}_j) + 2\epsilon$, *we have* $\hat{S}(\mathbf{x}_i) > \hat{S}(\mathbf{x}_j)$. *Therefore, the principle-based ordering is preserved up to a* $2\epsilon$ *margin. Furthermore, let* $\mathbf{x}^\star = \arg\max_n S^\star(\mathbf{x}_n)$ *and* $\hat{\mathbf{x}} = \arg\max_n \hat{S}(\mathbf{x}_n)$. *Then, the instantaneous regret of scheduling scores satisfies* $S^\star(\mathbf{x}^\star) - S^\star(\hat{\mathbf{x}}) \leq 2\epsilon$.

*Table 1.* Comparison of learned representations across scheduling approaches.

| APPROACH | LEARNED REPRESENTATION | TRANSFERABILITY | INTERPRETABILITY |
|---|---|---|---|
| INDEX POLICIES | ANALYTICALLY DEFINED INDEX | LIMITED | HIGH |
| INDEX LEARNING | APPROXIMATED INDEX | LIMITED | LOW–MEDIUM |
| RL SCHEDULING | POLICY OR $Q(s,a)$ | NO | LOW |
| STRUCTURED/ADDITIVE RL | ADDITIVE POLICY OR VALUE | NO | MEDIUM–HIGH |
| **FSP (OURS)** | PRIORITY PRINCIPLE $S(x)$ | EXPLICITLY CONSIDERED | HIGH |

**Transferability.** A key goal of the FSP framework is to generalize across different system characteristics (e.g., candidate sets with different sizes or item identities and different marginal feature distributions) under a shared supervision mechanism (e.g., reward structures or scheduling strategies). The principle invariance in Theorem 3.4 shows that a shared conditional supervision mechanism ensures that the learned scheduling principle is invariant to system characteristics.

**Theorem 3.4** (Principle invariance). *Let $Y$ denote the supervision signal. Suppose two systems $A$ and $B$ have different system characteristics, but share the same conditional supervision mechanism, i.e., for all $\mathbf{x} \in \mathcal{X}$, $\mathbb{P}_A(Y|X = \mathbf{x}) = \mathbb{P}_B(Y|X = \mathbf{x})$. If the loss function is strictly convex, then the unique minimizer $S^\star$ of the expected loss (population risk) between $S$ and $Y$ coincides across systems $A$ and $B$, and depends only on the conditional law $Y|X$.*

## 4. Related Work

Classical index-based scheduling originates from the multi-armed bandit literature. The Gittins index provides an optimal priority rule for rested bandits by assigning a scalar index to each arm based solely on its own state (Gittins et al., 2011; Scully & Terenin, 2025). The Whittle index extended this idea to restless bandits via Lagrangian relaxation to derive heuristic index policies (Whittle, 1988). Despite lacking general optimality guarantees, it has been widely applied in stochastic scheduling (Liu & Zhao, 2010; Maatouk et al., 2020). To relax the analytical requirements of closed-form indices, subsequent work has explored learning or approximating index functions from data (Fu et al., 2019; Xiong & Li, 2023; Robledo Relaño et al., 2024). While these methods preserve the interpretability of index-based rules, they still rely on indexability assumptions or treat the learned function as a direct proxy for an explicit index.

Learning-based scheduling has been widely studied through RL, where policies or value functions are optimized directly from experience data. Such approaches have demonstrated strong empirical performance in various domains, including recommender systems, wireless scheduling, and job-shop environments (Xu et al., 2017; Wei et al., 2018; Ye et al., 2019; Huang et al., 2021). However, RL-based schedulers typically operate on joint state representations and learn black-box policies or $Q$-functions, making the learned deci-

sion logic difficult to interpret and sensitive to changes in candidate sets (Lee et al., 2019; Milani et al., 2024).

To address these limitations, several works incorporate structured and additive models, such as generalized additive models (Hastie, 2017) and neural additive models (Agarwal et al., 2021), into learning-based scheduling to improve transparency while retaining learning-based optimization. Similar ideas have been applied to sequential decision problems by decomposing value or action-value functions to reveal feature-level effects in learned policies (Siems et al., 2023; Danach et al., 2025). Despite improved interpretability, these approaches typically treat the learned structure as a proxy for value estimation or embed it within policy optimization. As a result, they do not explicitly define or learn a transferable scheduling principle that can be applied independently of the policy or value representation. In contrast, our FSP framework explicitly learns a global priority principle $S(x)$ and enforces desired temporal properties. Table 1 summarizes these distinctions.

## 5. Experiments

### 5.1. Synthetic Experiment

We first validate the proposed framework on a controlled synthetic environment to examine whether the learning procedure can recover a known global scheduling principle. We consider a scheduling system with 8 candidate items (i.e., $N = 8$), with a four-dimensional feature space $\mathbf{x} = (x_1, x_2, x_3, x_4) \in [0,1]^4$. At each time step $t$, the environment presents the features of item $n$ $\{\mathbf{x}_n^t\}$ drawn i.i.d. from the uniform distribution on $[0,1]^4$. We define a synthetic global scheduling principle

$$
\begin{aligned}
S^\star(\mathbf{x}) = {} & 2\sin(x_1\pi) + 0.5(x_2 - 0.5)^2 + \\
& 1.5\max(x_3 - 0.3, 0) + 0.2 - 0.8(x_4 - 0.5)^2 + \\
& 0.6\sin(2x_1\pi)(x_2 - 0.5).
\end{aligned}
$$

This additive form captures nonlinear marginal effects of each feature and an interaction term of $x_1$ and $x_2$. When the policy selects item $n^t$ with feature $\mathbf{x}_{n^t}^t$, the reward is generated as $r^t = S^\star(\mathbf{x}_{n^t}^t) + \epsilon^t$, where $\epsilon^t \sim \mathcal{N}(0, \sigma^2)$. This ensures that the expected reward is aligned with the true principle, while introducing stochasticity.

The FSP framework parameterizes $S(\mathbf{x})$ as a structured additive function using spline bases for each coordinate.

The scheduling principle $\hat{S}$ is learned, i.e., the parameters are updated, by minimizing the combined likelihood and value regression loss in Eq. (7). We use the averaged state-value $V(\bar{s}^t) = \frac{1}{N^t} \sum_n S(\mathbf{x}_n^t)$ to stabilize training.

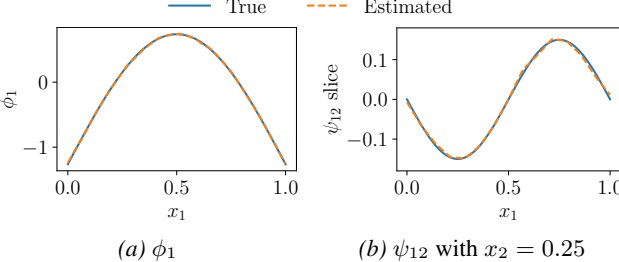

*(a) $\phi_1$*  *(b) $\psi_{12}$ with $x_2 = 0.25$*

*Figure 2.* Visual comparison of the component functions of $\hat{S}$ with the true principle $S^\star$.

First, we evaluate how $\hat{S}$ closely approximates the true principle $S^\star$. The function recovery metric, defined as the $L^2$ distance between $\hat{S}$ and $S^\star$ over a dense grid in $[0,1]^4$, is 0.013. This implies that the approximation of $S^\star$ is quite accurate. This is also shown clearly in Figure 2, which provides the visual comparison of the recovered principle, $\hat{\phi}_1$ and $\hat{\psi}_{12}$ with $x_2 = 0.25$, with the true principle. From the figure, we can see that the learned $\hat{S}$ closely follows the true principle $S^\star$. More visual comparisons of the recovered principle are provided in Appendix C.

*Table 2.* Transfer performance with varying $N$, where $\hat{S}$ is trained with $N = 8$.

| $N$ | RANKING CONSISTENCY | AVG. REWARD | REWARD GAP TO ORACLE |
|---|---|---|---|
| 4 | 0.9940 | 2.5262 | $7.9 \times 10^{-5}$ |
| 8 | 0.9947 | 2.7783 | $1.76 \times 10^{-4}$ |
| 16 | 0.9942 | 2.9377 | $2.37 \times 10^{-4}$ |
| 32 | 0.9941 | 3.0558 | $6.15 \times 10^{-4}$ |

We provide the ranking consistency and scheduling performance while varying the number of candidate items, $N$, in Table 2. Ranking consistency is assessed by the proportion of pairs $(i, j)$ for which $\text{sign}(S^\star(\mathbf{x}_i) - S^\star(\mathbf{x}_j)) = \text{sign}(\hat{S}(\mathbf{x}_i) - \hat{S}(\mathbf{x}_j))$. Across all settings of $N$, the learned policy achieves very close performance to the oracle, with average reward gaps on the order of $10^{-4}$. This indicates that the FSP framework is capable of approximating the optimal policy with very high fidelity. Moreover, the item ranking accuracy remains consistently around 99.4%, confirming that the learned value functions preserve the relative ordering of states almost perfectly. As $N$ increases, the average reward increases as expected due to the item diversity, yet the discrepancy between the learned and oracle policies remains negligible. This result demonstrates that the factorized representation generalizes robustly under transfer and scales reliably with larger environments.

This synthetic experiment directly verifies the identifiability and learnability of the global scheduling principle. The successful recovery of $S^\star$ shown in the results provides evidence that the FSP framework can discover interpretable and transferable principles that guide scheduling.

### 5.2. Realistic Scheduling Experiment

To validate the interpretability and robustness of the learned scheduling principles, we evaluate our framework on three realistic item-selection tasks with distinct structural characteristics: wireless user scheduling, inventory replenishment, and warehouse clearance. For each task, we use a four-dimensional feature vector $\mathbf{x}_n \in [0,1]^4$, which includes an intentionally uninformative random feature. This allows us to examine whether the learned principle emphasizes task-relevant conditions while suppressing spurious dimensions. The three tasks share a common single-item selection form, but differ in their coupling structure and penalty sensitivity. The detailed specifications for the realistic scheduling tasks are provided in Appendix D.

The selection of these tasks is motivated by the need to examine the transferability of scheduling policies. The wireless user scheduling task represents a system in which item selection is strongly coupled to item configurations. This is because the agent's decision for one user significantly affects others through shared penalties. In contrast, the inventory replenishment task provides a quasi-additive environment, where the system dynamics are relatively more localized. This approximate additivity can make item priorities more stable across feature-distribution changes for a fixed system size $N$. The warehouse clearance task complements these cases by introducing shared clearance capacity and severe overflow penalties, so poor rankings can cause large negative rewards. By comparing these tasks, we evaluate whether FSP can retain transferability across different values of $N$, even in highly coupled or penalty-sensitive settings, while achieving comparable levels of generalization within $N$ in more localized environments, thereby testing whether a reusable priority rule transfers across changes in system size and feature distribution.

We first evaluate the performance and transferability of the FSP framework on each realistic task. As a baseline, we consider the Whittle-index (Whittle, 1988), deep Q-network (DQN) (Mnih et al., 2015), advantage actor-critic (A2C) (Mnih et al., 2016), proximal policy optimization (PPO) (Schulman et al., 2017), trust region policy optimization (TRPO) (Schulman et al., 2015), quantile regression DQN (QR-DQN) (Dabney et al., 2018), and neural additive model (NAM) (Agarwal et al., 2021) trained on the same environments. Following Table 1, we use the Whittle-index as an index-policy baseline for transfer comparison. DQN, A2C, PPO, TRPO, QR-DQN, and NAM are included as fixed-size

*Table 3.* In-distribution reward, $10^{\text{ID}}$, and out-of-distribution average rewards under varying $N$ and system populations. FSP and Whittle are learned on the $10^{\text{ID}}$ system and transferred without retraining; the other baselines are trained for each $N$.

| TASK | $N$ | FSP | WHITTLE | NAM | DQN | A2C | PPO | TRPO | QR-DQN |
|------|-----|-----|---------|-----|-----|-----|-----|------|--------|
| | $10^{\text{ID}}$ | 0.109 | 0.114 | 0.129 | 0.112 | 0.094 | 0.107 | 0.112 | 0.127 |
| WIRELESS | 5 | 0.085±0.001 | 0.096±0.000 | 0.131±0.000 | -0.073±0.002 | 0.127±0.000 | 0.129±0.000 | 0.130±0.000 | 0.126±0.000 |
| | 10 | 0.103±0.001 | 0.115±0.001 | 0.109±0.001 | -1.313±0.003 | 0.095±0.001 | 0.104±0.001 | 0.112±0.001 | 0.119±0.001 |
| | 15 | -0.285±0.002 | -0.337±0.002 | -0.251±0.001 | -1.965±0.002 | -0.426±0.002 | -0.372±0.002 | -0.368±0.002 | -0.386±0.002 |
| | 20 | -0.781±0.001 | -0.888±0.001 | -0.752±0.001 | -2.609±0.001 | -1.045±0.002 | -0.971±0.002 | -0.951±0.002 | -1.019±0.002 |
| | $10^{\text{ID}}$ | -0.353 | -0.344 | -0.361 | -0.355 | -0.479 | -0.566 | -0.566 | -0.487 |
| INVENTORY | 5 | -0.206±0.011 | -0.231±0.010 | -0.232±0.014 | -0.228±0.012 | -0.157±0.017 | -0.137±0.016 | -0.142±0.018 | -0.127±0.015 |
| | 10 | -0.412±0.008 | -0.405±0.009 | -0.456±0.018 | -0.429±0.008 | -0.520±0.018 | -0.554±0.021 | -0.575±0.019 | -0.503±0.013 |
| | 15 | -0.712±0.008 | -0.723±0.008 | -0.763±0.017 | -0.727±0.008 | -0.701±0.023 | -0.700±0.020 | -0.679±0.017 | -0.643±0.012 |
| | 20 | -0.726±0.008 | -0.921±0.011 | -0.849±0.022 | -0.742±0.008 | -1.047±0.027 | -1.052±0.027 | -1.022±0.024 | -0.974±0.015 |
| | $10^{\text{ID}}$ | 0.823 | 0.737 | 0.446 | 0.883 | 0.824 | 0.866 | 0.826 | 0.884 |
| WAREHOUSE | 5 | 0.630±0.021 | 0.596±0.021 | -9.272±0.920 | -10.913±1.374 | -1.677±0.714 | -9.587±1.041 | -4.789±1.086 | -0.610±0.289 |
| | 10 | 0.738±0.019 | 0.677±0.019 | -18.031±1.269 | -35.238±1.931 | 0.682±0.021 | -7.817±1.470 | 0.696±0.021 | -1.974±0.457 |
| | 15 | 0.712±0.014 | 0.621±0.014 | -25.995±2.208 | -16.747±2.807 | -68.798±0.030 | -0.931±0.375 | 0.601±0.015 | 0.637±0.015 |
| | 20 | 0.649±0.023 | 0.605±0.015 | -30.154±2.545 | -9.063±1.913 | -92.641±0.035 | -83.264±0.118 | 0.545±0.015 | -63.028±0.412 |

learning baselines, highlighting the structural limitations of DRL policies or additive models whose representations are tied to a fixed candidate set size.

For each task, we train the FSP policy and estimate the Whittle-index only once on a training system with $N = 10$. Then, to demonstrate transferability, we apply FSP and Whittle-index policies to environments with $N \in \{5, 10, 15, 20\}$ and under different feature distributions, without any retraining, in a zero-shot manner. In contrast, other learning-based policies are trained individually for each system size $N$ due to their lack of transferability.

In Table 3, we provide both the in-distribution (ID) result on the training system and the out-of-distribution (OOD) transfer results across varying $N$ and system populations (i.e., variations in the distribution of item features). The OOD transfer results are averaged over 100 instances and reported with a 95% confidence interval. The comparison of the ID performance, $10^{\text{ID}}$, validates the scheduling performance of the FSP policy. Across all three tasks, the FSP policy achieves competitive performance relative to baseline policies. These results suggest that, even in realistic environments without a guaranteed global scheduling principle, a reusable and interpretable principle can be learned that generalizes across system conditions.

The OOD performance across $N \in \{5, 10, 15, 20\}$ evaluates different generalization settings for different methods. For FSP and the Whittle-index policy, the OOD results measure zero-shot transfer across both candidate set sizes and system populations. For the fixed-size learning baselines, the results measure generalization to different system populations at a fixed $N$.

For the wireless user scheduling task, the strongly coupled item configurations make transfer more difficult as the candidate set size increases. Despite relying on a single schedul-

ing principle learned on the $10^{\text{ID}}$ system, the FSP policy becomes increasingly competitive at larger values of $N$. The Whittle-index policy performs better than the FSP policy at $N = 5$ and $N = 10$, but the FSP policy overtakes it at $N = 15$ and $N = 20$. This suggests that a purely item-wise index approximation becomes less effective when system-level coupling effects become stronger. Among the fixed-size learning baselines, the NAM policy is one of the strongest methods and achieves the best reward in most OOD settings, suggesting that an additive value representation can be effective in this task. However, this result does not indicate zero-shot transfer across system sizes, because the NAM policy is trained separately for each value of $N$. Several other separately trained baselines are also stronger than FSP in smaller systems. In contrast, several fixed-size DRL policies, including DQN, A2C, PPO, TRPO, and QR-DQN, deteriorate substantially from $N = 15$ onward despite being trained separately for each system size. This indicates that strong per-size training does not necessarily yield robust population generalization, nor does it imply a reusable scheduling principle across larger systems.

For the inventory replenishment task, the localized and quasi-additive dynamics make per-size learning baselines competitive in selected regimes. Several DRL policies other than the DQN policy outperform the FSP policy at $N = 5$ and $N = 15$, showing that fixed-size learning baselines can exploit the relatively weak coupling when they are trained separately for each system size. However, this advantage is not consistent across $N$, as the FSP policy overtakes these DRL policies at $N = 10$ and becomes the best-performing method at $N = 20$ despite being trained only on the $10^{\text{ID}}$ system. The Whittle-index policy only slightly outperforms the FSP policy at the training system size $N = 10$ and falls behind at the other tested system sizes. Thus, although per-size training can be competitive in this quasi-additive environment, the FSP policy exhibits more stable transfer

as a single scheduling principle and becomes particularly strong at the largest tested system size.

For the warehouse clearance task, distribution shifts can cause costly scheduling ranking errors, which may result in severe overflow penalties. Many fixed-size learning policies therefore incur large negative rewards under OOD evaluation. The NAM and DQN policies fail across all tested system sizes, while A2C, PPO, and QR-DQN also deteriorate at several values of $N$. In contrast, the FSP policy achieves the highest reward across all tested system sizes despite being trained only on the $10^{\text{ID}}$ system. The Whittle-index policy is more stable than most fixed-size learning policies, but it remains below the FSP policy for every tested system size. This indicates that the FSP policy is the most robust method in this task, maintaining stable zero-shot transfer under both system-size and population shifts.

These OOD results over different tasks show that FSP does not merely fit the training system, but learns a reusable scheduling principle that remains robust under changes in system size and heterogeneous item populations.

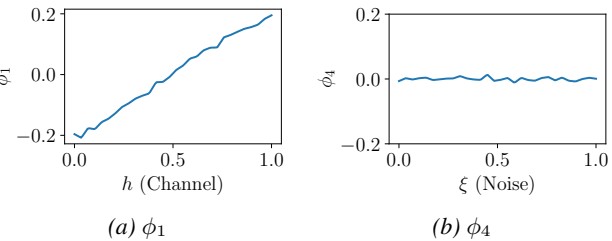

*(a) $\phi_1$*      *(b) $\phi_4$*

*Figure 3.* Two of four one-dimensional component functions of the FSP policy in the wireless user scheduling task.

To show the interpretability of the FSP framework, Figure 3 illustrates the selected additive components from the learned principle for the wireless user scheduling task. In Figure 3a, the scoring function $\phi_1(h)$ associated with the channel condition is monotonically increasing, indicating that users with better channel quality are consistently assigned higher priority. This aligns with domain intuition, as improved channel conditions directly lead to higher achievable rewards. In contrast, the scoring function $\phi_4(\xi)$ corresponding to the intentionally uninformative noise feature remains nearly flat, confirming that the learned scheduling principle correctly assigns negligible importance to irrelevant information. These results demonstrate that the FSP framework learns interpretable prioritization rules that reflect meaningful domain factors, while suppressing spurious correlations. This transparency provides insight into the learned decision logic and helps explain why the same priority rule can be reused across system instances.

Additional results and more detailed analyses of the realistic experiments, including reward performance as well as additive one-dimensional and two-dimensional pairwise components, are provided in Appendix E.

## 6. Discussions

**Towards more practical scheduling.** Practical systems often involve additional system factors, sub-decisions beyond item selection, and multi-item decisions. Such factors can be incorporated through additional additive system terms or item-system interaction components without sacrificing interpretability. Moreover, the principle can be extended to handle sub-decisions and multi-item selection by introducing structured residual components and considering the learned scores as reusable primitives for combinatorial settings. These observations suggest that the proposed framework naturally extends to more realistic and complex scheduling scenarios.

**Scalability.** Although FSP is designed to be reusable across different candidate set sizes, scalability remains a limitation in high-dimensional feature spaces. In particular, the number of possible pairwise interaction components grows quadratically with the number of features, so parameterizing all interaction components may increase computational cost and reduce interpretability. While our experiments demonstrate strong scalability with respect to the number of candidate items $N$, this issue may become more important in high-dimensional scheduling systems. Therefore, applying FSP to larger feature spaces may require sparse interaction selection, low-rank interaction structures, or domain-guided restrictions on admissible interactions.

**Assumption on the existence of a global principle.** Our theoretical analysis assumes the existence of a global scheduling principle $S^\star$ that determines rewards as a function of feature conditions. This assumption enables identifiability and regret guarantees, but may be restrictive in settings where no exact additive structure exists. In such cases, the proposed framework should be viewed as learning an approximate yet interpretable surrogate principle. Nevertheless, realistic scheduling experiments reveal that the learned principle remains informative and robust even when the system does not strictly follow an additive structure.

## 7. Conclusion

In this paper, we proposed the FSP framework, which learns an interpretable scheduling priority rule directly from experience. It preserves a simple, priority-based factorized structure while achieving robust performance and transferability across different system scales and feature distributions. Through experiments on both synthetic and realistic tasks, we demonstrated that the framework can learn reusable and interpretable scheduling principles that generalize beyond the training system, even when an exact global principle is not guaranteed. These results show that the proposed framework provides a practical approach for learning interpretable and transferable scheduling rules.

## Acknowledgements

This work was supported in part by the National Research Foundation of Korea (NRF) grant funded by the Korea government (MSIT) (RS-2025-24523498 and RS-2026-25482257) and in part by the IITP(Institute of Information & Communications Technology Planning & Evaluation)-ITRC(Information Technology Research Center) grant funded by the Korea government (Ministry of Science and ICT) (IITP-2026-RS-2021-II211816). We thank all reviewers for their valuable comments and suggestions.

## Impact Statement

This paper presents work whose goal is to advance the field of machine learning, with a focus on learning interpretable and transferable scheduling principles. The proposed framework is intended to improve transparency and robustness of decision-making systems, and does not introduce new ethical risks beyond those typically associated with learning-based optimization methods. We do not anticipate any negative societal impacts from this work in the near future.

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

## A. Algorithmic Details

This appendix provides the learning and inference algorithms for the global scheduling principle $S(x)$. The learning algorithm updates the parameters of the structured additive model $\{\phi_k, \psi_{kl}\}$, $\Theta$, using both likelihood-based and value-regression losses, while enforcing regularization and identifiability constraints. The inference algorithm shows how the learned principle is used at deployment to score and select items.

---

**Algorithm 1** Learning the Global Scheduling Principle $S(x)$

---

**Require:** Feature space $[0,1]^K$, temperature $\tau > 0$, discount factor $\gamma \in [0,1]$, trade-off hyperparameter $\lambda \geq 0$, learning rate $\eta$

1: **Model:** $S(\mathbf{x}; \Theta) = \sum_{k=1}^{K} \phi_k(x_k) + \sum_{(k,l)\in\mathcal{P}} \psi_{kl}(x_k, x_l)$
2: **Identifiability:** Impose the constraints in Eq. (2)–(5).
3: Initialize parameters $\Theta \leftarrow \{\phi_k\}_{k=1}^{K} \cup \{\psi_{kl}\}_{(k,l)\in\mathcal{P}}$
4: **for** $t = 1, 2, \cdots$ **do**
5:     Observe candidate set at time step $t$, $\mathcal{N}(t) = \{\mathbf{x}_n^t\}_{n=1}^{N^t}$
6:     Compute item scores $s_n \leftarrow S(\mathbf{x}_n^t; \Theta)$ for all $n \in \mathcal{N}(t)$
7:     Obtain the softmax policy $\pi(n \mid \mathcal{N}(t)) \leftarrow \frac{\exp(s_n/\tau)}{\sum_{m=1}^{N^t} \exp(s_m/\tau)}$
8:     Sample action $n^t \sim \pi(\cdot \mid \mathcal{N}(t))$, execute, observe reward $r^t$ and next candidate set $\mathcal{N}(t{+}1)$
9:     Calculate likelihood loss as $\mathcal{L}_{\text{lik}} \leftarrow -r^t \log \pi(n^t \mid \mathcal{N}(t))$
10:    Calculate state values as

$$V(\bar{s}^t) \leftarrow \frac{1}{N^t} \sum_{n=1}^{N^t} S(\mathbf{x}_n^t; \Theta) \text{ and } V(\bar{s}^{t+1}) \leftarrow \frac{1}{N^{t+1}} \sum_{n=1}^{N^{t+1}} S(\mathbf{x}_n^{t+1}; \Theta)$$

11:    Calculate value loss as $\mathcal{L}_{\text{val}} \leftarrow (y^t - V(\bar{s}^t))^2$, where the TD target is given by $y^t \leftarrow r^t + \gamma V(\bar{s}^{t+1})$
12:    Calculate regularization loss as $\mathcal{R} \leftarrow \Omega(\phi, \psi)$ (e.g., $l_2$ penalty)
13:    Obtain total loss as $\mathcal{L} \leftarrow \mathcal{L}_{\text{lik}} + \lambda \mathcal{L}_{\text{val}} + \mathcal{R}$
14:    Update parameters using the total loss as $\Theta \leftarrow \Theta - \eta \nabla_\Theta \mathcal{L}$
15:    Project the parameters $\Theta$ to the identifiability set
16: **end for**

---

**Algorithm 2** Inference / Deployment (Scoring and Scheduling)

---

**Require:** Learned $S(\mathbf{x}; \hat{\Theta}) = \sum_k \phi_k(x_k) + \sum_{(k,l)\in\mathcal{P}} \psi_{kl}(x_k, x_l)$

1: Observe candidate set $\mathcal{N}(t) = \{\mathbf{x}_n^t\}_{n=1}^{N^t}$
2: Compute scores $s_n \leftarrow S(\mathbf{x}_n^t; \hat{\Theta})$ for all $n \in \mathcal{N}^t$
3: Select item $n^t \leftarrow \arg\max_{n \in \mathcal{N}^t} s_n$
4: Execute action $n^t$

---

## B. Assumptions and Proofs of Theoretical Results

### B.1. Assumptions

We summarize the standing assumptions used in the theoretical results. We only mention assumptions in proofs only if they are significant.

(A1) **Existence of global principle.** There exists a (possibly unknown) $S^\star : [0,1]^K \to \mathbb{R}$ that determines the expected reward of selecting an item with feature $x$, i.e., $\mathbb{E}[r^t \mid \mathbf{x}_n^t = x] = g(S^\star(x))$ for a nondecreasing $g$, which is $L_g$-Lipschitz and bounded in $[0,1]$.

(A2) **Single-item selection, weak interaction** At each step one item is selected. The (immediate) reward depends on the selected item's feature through $S^\star$; unchosen items affect reward only via constraints/normalization (no direct multi-item payoff).

(A3) **Sampling for learning/generalization** For approximation and generalization results, we assume i.i.d. sampling of feature points from a stationary distribution $P$ on $[0,1]^K$.

(A4) **Regularity** $S^\star$ is $L$-Lipschitz and its univariate (bivariate) components admit bounded second mixed derivatives; our bases are given by splines/lattices and sufficiently dense.

### B.2. Proof of Theorem 3.1 (Identifiability)

*Proof.* Suppose $S$ has two admissible decompositions $(\{\phi_k\}, \{\psi_{kl}\})$ and $(\{\tilde{\phi}_k\}, \{\tilde{\psi}_{kl}\})$ satisfying the centering constraints. Let

$$\Delta(\mathbf{x}) = \sum_{k=1}^K \left(\phi_k(x_k) - \tilde{\phi}_k(x_k)\right) + \sum_{(k,l)\in\mathcal{P}} \left(\psi_{kl}(x_k, x_l) - \tilde{\psi}_{kl}(x_k, x_l)\right),$$

so that $\Delta(\mathbf{x}) \equiv 0$. We define $\delta\phi_k := \phi_k - \tilde{\phi}_k$ and $\delta\psi_{kl} := \psi_{kl} - \tilde{\psi}_{kl}$. For a function $f$ on $[0,1]^K$ and an index set $I \subseteq [K] = \{1, \ldots, K\}$, define the marginal projection on the coordinates $I$ as

$$(\Pi_I f)(\mathbf{x}_I) := \int_{[0,1]^{K-|I|}} f(\mathbf{x}) \, d\mathbf{x}_{[K]\setminus I},$$

where $\mathbf{x}_I = \{x_i : i \in I\}$ and $d\mathbf{x}_{[K]\setminus I}$ denotes integration with respect to all coordinates except those in $I$. First, we apply the projection operator, $\Pi_{\{k\}}$, to $\Delta$:

$$0 = (\Pi_{\{k\}}\Delta)(x_k) = \delta\phi_k(x_k) + \sum_{j\neq k} \int_0^1 \delta\phi_j(x_j) \, dx_j + \sum_{(i,j)\in\mathcal{P}} (\Pi_{\{k\}}\delta\psi_{ij})(x_k).$$

The second term vanishes by the mean centering constraint in Equation (2). For each $(i,j) \in \mathcal{P}$, $(\Pi_{\{k\}}\delta\psi_{ij})$ vanishes by the row/column and global centering constraints in Equations (3)–(5). If $i = k$ or $j = k$, the row/column centering constraints yield 0; if $\{i,j\} \cap \{k\} = \emptyset$, both $x_i, x_j$ are integrated out and the global centering constraint yields 0. Repeating this for all $k$ yields that, for all $k$, $\phi_k(x_k) = \tilde{\phi}_k(x_k)$ a.e. Then, $\Delta$ becomes:

$$\Delta(\mathbf{x}) = \sum_{(k,l)\in\mathcal{P}} \delta\psi_{kl}(x_k, x_l).$$

We fix $(k,l) \in \mathcal{P}$ and apply $\Pi_{\{k,l\}}$ to $\Delta$:

$$0 = (\Pi_{\{k,l\}}\Delta)(x_k, x_l) = \delta\psi_{kl}(x_k, x_l) + \sum_{(i,j)\in\mathcal{P}\setminus\{(k,l)\}} (\Pi_{\{k,l\}}\delta\psi_{ij})(x_k, x_l).$$

For $(i,j) \neq (k,l)$, $(\Pi_{\{k,l\}}\delta\psi_{ij})$ vanishes by: if exactly one of $\{i,j\}$ lies in $\{k,l\}$, the remaining coordinate is integrated out and the row/column centering constraints yield 0; if $\{i,j\} \cap \{k,l\} = \emptyset$, both $x_i, x_j$ are integrated out and the global centering constraint yields 0. Therefore, $\delta\psi_{kl} \equiv 0$ a.e. for all $(k,l) \in \mathcal{P}$. Consequently, $\phi_k = \tilde{\phi}_k$ and $\psi_{kl} = \tilde{\psi}_{kl}$ a.e., proving uniqueness. $\qquad\square$

### B.3. Proof of Theorem 3.2 (Approximation Error)

With Assumption A4, we can show the theorem by directly following standard spline approximation theory (Schumaker, 2007). For univariate components, cubic spline interpolation achieves $\|f - f_h\|_\infty = O(h^2)$ for $C^2$ smooth functions, where $h = 1/R$. For bivariate components, tensor-product splines achieve the same $O(h^2)$ rate under $C^{2,2}$ smoothness. Combining these bounds across additive components yields the stated result.

### B.4. Proof of Theorem 3.3 (Ranking Consistency & Greedy regret)

*Proof.* By the assumptions, $|S^\star(\mathbf{x}) - \hat{S}(\mathbf{x})| \leq \epsilon$ for all $\mathbf{x}$. Then, for any items $\mathbf{x}_i, \mathbf{x}_j$, we have

$$\hat{S}(\mathbf{x}_i) - \hat{S}(\mathbf{x}_j) \geq S^\star(\mathbf{x}_i) - \epsilon - (S^\star(\mathbf{x}_j) + \epsilon) = (S^\star(\mathbf{x}_i) - S^\star(\mathbf{x}_j)) - 2\epsilon.$$

Therefore, if $S^\star(\mathbf{x}_i) - S^\star(\mathbf{x}_j) > 2\epsilon$, then $\hat{S}(\mathbf{x}_i) > \hat{S}(\mathbf{x}_j)$, which proves the consistency of the ranking. Additionally, we have

$$S^\star(\hat{\mathbf{x}}) \geq \hat{S}(\hat{\mathbf{x}}) - \epsilon \geq \hat{S}(\mathbf{x}^\star) - \epsilon \geq S^\star(\mathbf{x}^\star) - 2\epsilon,$$

where the second inequality comes from the definition of $\hat{\mathbf{x}}$. Thus, the regret $S^\star(\mathbf{x}^\star) - S^\star(\hat{\mathbf{x}})$ is at most $2\epsilon$. $\qquad\square$

### B.5. Proof of Theorem 3.4 (Principle invariance)

*Proof.* Define the hypothesis class $\mathcal{S} = \{S : \mathcal{X} \to \mathbb{R} \mid S \text{ additive with centering}\}$. Let the population risk be $\mathcal{R}_P(S) = \mathbb{E}_{(X,Y)\sim P}\big[\ell(S(X), Y)\big]$. For each $\mathbf{x} \in \mathcal{X}$ and scalar $z \in \mathbb{R}$, define the conditional risk

$$L_{\mathbf{x}}(z) := \mathbb{E}[\ell(z, Y)|X = \mathbf{x}].$$

By strict convexity of $\ell(\cdot, y)$, $L_{\mathbf{x}}(z)$ admits a unique minimizer $z^\star(x)$. For any marginal $P(X)$, the population risk decomposes as

$$\mathcal{R}_P(S) = \mathbb{E}_{X\sim P}\big[L_X(S(X))\big],$$

which is minimized pointwise at $S^\star(\mathbf{x}) = z^\star(\mathbf{x})$ for each $\mathbf{x}$, independently of $P(X)$. Therefore, if two systems $A$ and $B$ share the same conditional law $Y|X$, they share the same minimizer $S^\star$, i.e., the optimal scheduling principle is invariant across systems $A$ and $B$. Identifiability constraints ensure uniqueness of the decomposition into $\{\phi_k, \psi_{kl}\}$. □

## C. Additional Results for Synthetic Experiments

This appendix provides additional visualizations that support the synthetic experiment reported in Section 5.1. The figures illustrate how accurately the proposed FSP framework recovers the underlying global scheduling principle by comparing the learned components with the ground-truth ones.

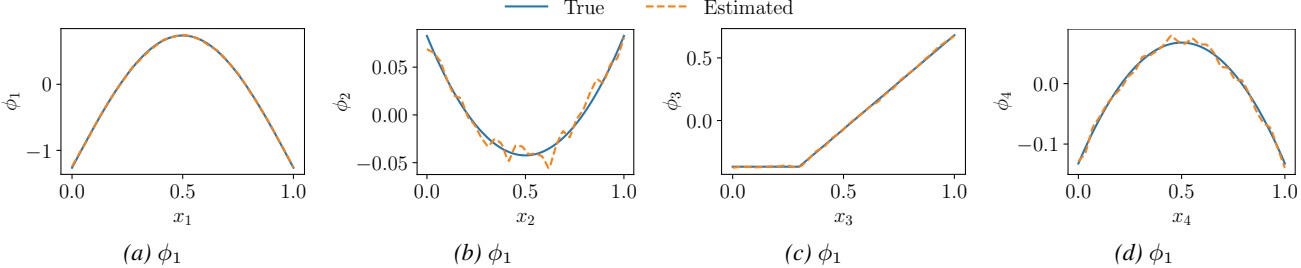

*(a) $\phi_1$*     *(b) $\phi_1$*     *(c) $\phi_1$*     *(d) $\phi_1$*

*Figure 4.* The visual comparison of the univariate component functions of $\hat{S}$ with the true principle $S^\star$.

Figure 4 visualizes the learned univariate component functions $\hat{\phi}_k$ together with the corresponding ground-truth components $\phi_k^\star$ for all feature dimensions. The close overlap between the estimated and true curves across the entire feature domain indicates that the proposed method accurately recovers the marginal effects of individual features, validating the effectiveness of the factorized representation and the imposed identifiability constraints.

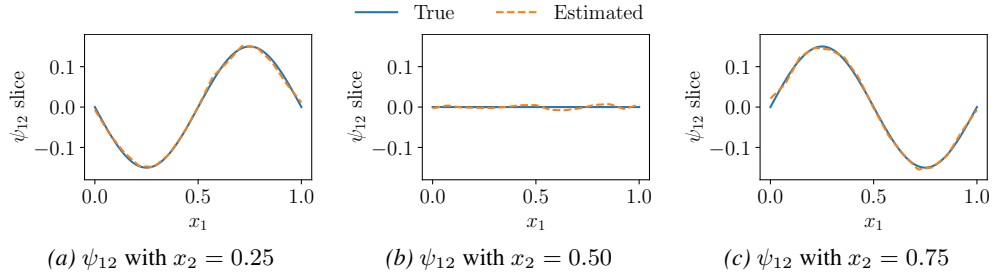

*(a) $\psi_{12}$ with $x_2 = 0.25$*     *(b) $\psi_{12}$ with $x_2 = 0.50$*     *(c) $\psi_{12}$ with $x_2 = 0.75$*

*Figure 5.* The one-dimensional visual comparison of the pairwise component function of $\hat{S}$ with the true principle $S^\star$.

Figure 5 presents one-dimensional slices of the learned pairwise interaction component function $\hat{\psi}_{12}$ at several fixed values of the second feature. These slices are compared against the corresponding ground-truth interactions and demonstrate that the learned interaction function closely matches the true structure across different conditioning values, capturing the non-additive dependence between features.

Figure 6 shows a two-dimensional surface visualization of the learned pairwise interaction component $\hat{\psi}_{12}$ alongside the ground-truth surface $\psi_{12}^\star$, as well as their absolute difference. The surface plots confirm that the learned interaction accurately

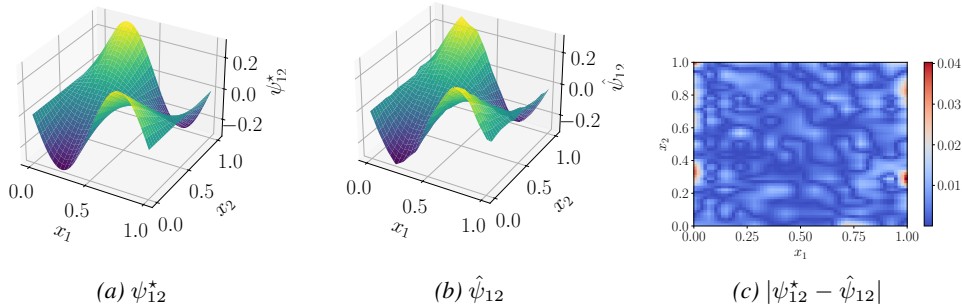

*(a) $\psi_{12}^{\star}$*              *(b) $\hat{\psi}_{12}$*              *(c) $|\psi_{12}^{\star} - \hat{\psi}_{12}|$*

*Figure 6.* The two-dimensional visual comparison of the component functions of $\hat{S}$ with the true principle $S^{\star}$.

reproduces the global shape and magnitude of the true interaction, while the error visualization indicates uniformly small discrepancies over the feature space.

Overall, these visualizations provide qualitative evidence that the proposed FSP framework can reliably recover both univariate and pairwise components of the global scheduling principle. Together with the quantitative results in Section 5.1, they demonstrate that the learned scheduling principles are interpretable and faithfully capture the underlying structure of the synthetic environment.

## D. Details of Realistic Scheduling Environments

This appendix provides the detailed specifications for realistic scheduling tasks, including ones used in the realistic scheduling experiments in Section 5.2. These tasks are based on the standard scheduling formulation introduced in Section 2.2, while incorporating stochastic features, constraints, and competing economic objectives. Therefore, the scheduling principles of these tasks are not trivial.

### D.1. Wireless User Scheduling

At each time step $t$, the wireless user scheduling environment presents a candidate set of $N$ users, where each user is characterized by a feature vector $\mathbf{x}_n^t \in [0,1]^4$:

$$\mathbf{x}_n^t = \left( h_n^t, \ \tau_n^t, \ q_n^t, \ \xi_n^t \right),$$

where $h_n^t$ represents the normalized channel quality (e.g., capacity), $\tau_n^t$ denotes the age of information (AoI) representing the time elapsed since the last successful service, $q_n^t$ is the normalized buffer occupancy (queue length), and $\xi_n^t$ is an uninformative random noise feature. At each step, the scheduler selects an action $a^t \in \{1, \ldots, N\}$ to grant exclusive access to the wireless channel. The transmission capacity for the selected user is determined by a non-linear interaction between the channel quality $h_{a^t}^t$ (scaled by $\eta$) and the available data in the buffer $q_{a^t}^t$. The realized throughput $T^t$ is limited by the available data in the buffer:

$$T^t = \min\left( \eta \cdot h_{a^t}^t, \ q_{a^t}^t \right).$$

The system dynamics evolve as follows. For the selected user $n = a^t$, the AoI is reset to zero, while for unselected users, it increases linearly:

$$\tau_n^{t+1} = \begin{cases} 0, & n = a^t, \\ \min\{\tau_n^t + \Delta\tau, \ 1\}, & n \neq a^t, \end{cases}$$

where $\Delta\tau$ is a fixed increment with noise. Buffer occupancy is updated by incorporating stochastic packet arrivals $\lambda_n^t$ and subtracting the transmitted data:

$$q_n^{t+1} = \min\left( \max\{q_n^t + \lambda_n^t - \mathbf{1}(n = a^t)T^t, \ 0\}, \ 1 \right),$$

where $\mathbf{1}(\cdot)$ is an indicator function. To simulate realistic bursty traffic, the arrival process $\lambda_n^t$ follows a Gamma distribution, $\lambda_n^t \sim \mathrm{Gamma}(k, \theta)$, truncated to preserve the normalized range. Any arrival exceeding the buffer capacity is recorded as an

overflow. The immediate reward $r^t$ aggregates throughput and system-wide penalties:

$$r^t = \omega T^t - \lambda_{\mathrm{ov}} \sum_{n=1}^{N} \max\{q_n^t + \bar{\lambda} - 1,\ 0\} - \lambda_{\mathrm{delay}} \bar{\tau}_{t+1},$$

where $\omega$ is the throughput weight, $\lambda_{\mathrm{ov}}$ is the overflow penalty coefficient, and $\lambda_{\mathrm{delay}}$ scales the delay penalty. The overflow penalty is a non-linear function based on a constant mean arrival rate $\bar{\lambda}$. The delay penalty is defined as the arithmetic mean of the AoI across all users after the reset of the scheduled user's delay:

$$\bar{\tau}_{t+1} = \frac{1}{N} \sum_{n=1}^{N} \tau_n^t - \frac{1}{N} \tau_{a^t}^t.$$

This environment presents a significant challenge to the scheduling agent due to the non-additive and non-linear nature of the penalties. In particular, it is characterized by strong global coupling and pronounced non-linearity, arising from the throughput constraint $T^t = \min(\eta h_{a^t}^t, q_{a^t}^t)$ and the system-wide AoI penalty. These effects are further amplified by the high variance induced by bursty Gamma-distributed traffic, resulting in a value function that is highly sensitive to the joint system state, where the relative priority of a user can vary substantially with the states of others. Consequently, this environment requires the agent to learn a prioritization principle that balances spectral efficiency with long-term system stability. The dependence of the penalties on aggregate system states makes this task a representative testbed in which the optimal scheduling policy deviates markedly from simple additive assumptions.

### D.2. Inventory Replenishment

At each time step $t$, the inventory replenishment environment presents a candidate set $\{\mathbf{x}_n^t\}_{n=1}^{N}$, where each item feature $\mathbf{x}_n^t \in [0, 1]^4$ follows the same convention as in the main text. Specifically, the feature vector is defined as

$$\mathbf{x}_n^t = \left( I_n^t,\ \mu_n^t,\ m_n^t,\ \xi_n^t \right),$$

where $I_n^t$ denotes the normalized inventory level, $\mu_n^t$ the demand intensity, $m_n^t$ the unit margin, and $\xi_n^t$ is an intentionally uninformative random feature. The random feature $\xi_n^t$ is included to examine whether the learned scheduling principle assigns negligible marginal effects to irrelevant dimensions. In a system, the demand intensity and unit margin of each item are based on fixed values, but time-varying with Gaussian noise of $\epsilon_{\mu,n}^t \sim \mathcal{N}(0, \sigma_\mu^2)$ and $\epsilon_{m,n}^t \sim \mathcal{N}(0, \sigma_m^2)$, respectively. The agent selects an action $a^t \in \{1, \ldots, N\}$ indicating the item to prioritize for replenishment. The environment enforces a single-item replenishment constraint. Given the selected action $a^t$, replenishment quantities are defined as

$$q_n^t = \begin{cases} Q, & n = a^t, \\ 0.5\,Q, & n \neq a^t, \end{cases}$$

where $Q > 0$ is a fixed replenishment amount. The post-replenishment inventory is updated as

$$\tilde{I}_n^t = \min\{I_n^t + q_n^t,\ 1\},$$

and any excess beyond capacity is recorded as overflow. After replenishment, stochastic demand is realized independently for each item. The demand for item $n$ is sampled according to

$$D_n^t \sim \mathrm{Binomial}(K_d,\ \mu_n^t)/K_d,$$

where $K_d$ controls the granularity of demand realization. The sales and next-step inventory are computed as

$$S_n^t = \min\{\tilde{I}_n^t,\ D_n^t\}, \qquad I_n^{t+1} = \tilde{I}_n^t - S_n^t,$$

and the stock-out is defined as $\max\{D_n^t - \tilde{I}_n^t,\ 0\}$. In particular, the unit margin $m_n$ affects the reward through the price term $p_n = c_n + m_n$, so that items with higher margins yield larger revenue. The immediate reward $r^t$ aggregates multiple economic components:

$$r^t = w_{\text{rev}} \sum_{n=1}^{N} p_n S_n^t - w_{\text{ord}} \sum_{n=1}^{N} c_n q_n^t - w_{\text{hold}} \sum_{n=1}^{N} h_n I_n^{t+1} - w_{\text{so}} \sum_{n=1}^{N} s_n \max\{D_n^t - \tilde{I}_n^t,\ 0\} - w_{\text{ov}}\lambda \sum_{n=1}^{N} \max\{I_n^t + q_n^t - 1,\ 0\},$$

where $p_n$, $c_n$, $h_n$, and $s_n$ denote the price, unit cost, holding cost, and stock-out penalty of item $n$, respectively. The scalar $\lambda$ controls the overflow penalty. This reward structure induces partially conflicting incentives and does not reduce to an additive function of the selected item's features.

This environment presents non-additive characteristics due to the unselected items influence the reward indirectly through shared replenishment constraints, inventory carry-over, and overflow penalties. Nevertheless, it exhibits a quasi-additive structure with relatively localized dynamics because unselected items still receive partial replenishment $(0.5Q)$ that serves as a buffer against state deterioration. The inherent additive reward structure allows the scheduling agent to learn a more consistent prioritization logic based on individual item features rather than complex global interdependencies. This predictable structure enables DQN to maintain a reasonable level of transferability within a fixed $N$, as the relative value of replenishing an item remains stable across different permutations.

### D.3. Warehouse Clearance

In addition to the two realistic tasks in Section 5.2, we introduce a warehouse clearance environment. This environment demonstrates that a lack of transferability in learned prioritization rules can directly lead to severe performance degradation, manifested as frequent overflow events and large reward failures.

At each time step $t$, the warehouse clearance environment consists of $N$ items represented by feature vectors $\{\mathbf{x}_n^t\}_{n=1}^{N}$, where $\mathbf{x}_n^t \in [0, 1]^4$ and

$$\mathbf{x}_n^t = \left(I_n^t,\ \rho_n^t,\ m_n^t,\ \xi_n^t\right).$$

Here, $I_n^t$ denotes the inventory level of item $n$, $\rho_n^t$ is the inventory inflow rate, $m_n^t$ is the unit margin, and $\xi_n^t$ is an intentionally uninformative random feature independently resampled at every time step. Each item is associated with fixed base inflow rate and base margin, which are perturbed by Gaussian noise at each time step, $\epsilon_{\rho,n}^t \sim \mathcal{N}(0, \sigma_\rho^2)$ and $\epsilon_{m,n}^t \sim \mathcal{N}(0, \sigma_m^2)$, respectively. These base values differ across items and induce persistent heterogeneity in inventory accumulation and profitability.

At each time step, the agent selects a single item $a^t \in \{1, \ldots, N\}$ to clear. Only the selected item generates sales, with realized sales quantity

$$S_{a^t}^t = \min\{I_{a^t}^t,\ C\},$$

where $C$ denotes a global sales capacity. The immediate revenue is given by

$$R^t = \alpha\, m_{a^t}^t\, S_{a^t}^t,$$

with $\alpha$ being a fixed scaling factor. Inventory evolves deterministically according to

$$I_n^{t+1} = \min\big\{\max\big\{I_n^t + \rho_n^t - \mathbb{I}\{n = a^t\}S_{a^t}^t, 0\big\}, 1\big\},$$

where inventories are strictly constrained by capacity. The reward at time $t$ is defined as

$$r^t = R^t - \sum_{n=1}^{N} h_n I_n^{t+1} - \sum_{n=1}^{N} \lambda\, \mathbb{I}\{I_n^{t+1} \geq \tau\},$$

where the final term imposes a large overflow penalty when an item's inventory exceeds a threshold $\tau$ close to capacity.

Although this problem appears similar to inventory replenishment at first, its failure mode is fundamentally different. DQN tightly couples action values tightly to item-level features, implicitly assuming that the relative desirability of an item remains stable across system configurations. However, in the presence of severe overflow penalties, this assumption breaks down. Then, items that are not selected accumulate inventory deterministically and can suddenly incur large negative rewards once capacity thresholds are exceeded. Consequently, policies learned by DQN overfit to system-specific overflow patterns rather than learning transferable prioritization rules. When the number of items or their ordering changes, the learned value function fails to anticipate which items are likely to trigger overflow. This leads to frequent capacity violations and sharply degraded performance. This strong coupling between item features and global overflow dynamics makes generalization particularly fragile in such environments.

### D.4. Experimental Settings

We summarize the experimental settings used for the realistic scheduling experiments, including common configurations shared across tasks as well as task-specific system parameters.

**Common Settings**    For the proposed FSP framework, the scheduling principle is parameterized using additive item-level components and pairwise interaction terms. The model is trained using stochastic gradient descent with a fixed learning rate. The Whittle-index baseline uses an item-wise index policy estimated from the training system. The NAM baseline replaces the Q-network in DQN with a neural additive model. For DQN-based baselines, a standard DQN algorithm is used to approximate the action-value function, taking the concatenated item features as input. A2C, PPO, TRPO, and QR-DQN use the same concatenated item-feature input as DQN and are implemented using Stable-Baselines and its contributed extensions. Unless otherwise specified, all policies are trained with $N = 10$ items and evaluated under both in-distribution and out-of-distribution settings.

*Table 4.* Hyperparameter summary for FSP and baseline methods.

| FSP | | WHITTLE-INDEX POLICY | |
|---|---|---|---|
| PARAMETER | VALUE | PARAMETER | VALUE |
| LEARNING RATE ($\eta$) | 0.005 | STATE DISCRETIZATION | $4 \times 4 \times 4$ |
| BASIS TYPE | B-SPLINE FUNCTIONS | DISCOUNT FACTOR | 0.99 |
| 1-D BASIS DIMENSION | 30 | DIRICHLET PRIOR | 0.3 |
| 2-D BASIS DIMENSION | 15 | VALUE-ITERATION THRESHOLD | $10^{-3}$ |
| TRADE-OFF HYPERPARAMETER ($\lambda$) | 0.1 | SUBSIDY-SEARCH THRESHOLD | $5 \times 10^{-2}$ |
| L2 REGULARIZATION | 0.0001 | | |

| NAM | | DQN | |
|---|---|---|---|
| PARAMETER | VALUE | PARAMETER | VALUE |
| Q-NETWORK TYPE | FEATURE-ADDITIVE | Q-NETWORK TYPE | MLP |
| ADDITIVE BLOCK DEPTH | 1 | Q-NETWORK ARCHITECTURE | (128,128) UNITS |
| LEARNING RATE | 0.001 | LEARNING RATE | 0.001 |
| DISCOUNT FACTOR | 0.99 | DISCOUNT FACTOR | 0.99 |
| BATCH SIZE | 256 | BATCH SIZE | 256 |
| REPLAY BUFFER | 20,000 | REPLAY BUFFER | 100,000 |

| A2C | | PPO | |
|---|---|---|---|
| PARAMETER | VALUE | PARAMETER | VALUE |
| ACTOR-CRITIC NETWORK TYPE | MLP | ACTOR-CRITIC NETWORK TYPE | MLP |
| ACTOR-CRITIC ARCHITECTURE | (128,128) UNITS | ACTOR-CRITIC ARCHITECTURE | (128,128) UNITS |
| LEARNING RATE | 0.001 | LEARNING RATE | 0.001 |
| DISCOUNT FACTOR | 0.99 | DISCOUNT FACTOR | 0.99 |
| | | BATCH SIZE | 64 |

| TRPO | | QR-DQN | |
|---|---|---|---|
| PARAMETER | VALUE | PARAMETER | VALUE |
| ACTOR-CRITIC NETWORK TYPE | MLP | Q-NETWORK TYPE | QUANTILE MLP |
| ACTOR-CRITIC ARCHITECTURE | (128,128) UNITS | Q-NETWORK ARCHITECTURE | (128,128) UNITS |
| LEARNING RATE | 0.001 | LEARNING RATE | 0.001 |
| DISCOUNT FACTOR | 0.99 | DISCOUNT FACTOR | 0.99 |
| TARGET KL | 0.01 | BATCH SIZE | 32 |
| | | REPLAY BUFFER | 1,000,000 |

**Wireless User Scheduling System Parameters**    The wireless scheduling environment is governed by a set of parameters that control traffic dynamics and performance trade-offs. Specifically, the mean arrival rate and the shape parameter determine the stochastic characteristics of user traffic arrivals. The capacity scaling parameter controls the effective service rate, while

a throughput weight balances the contribution of achieved throughput to the reward. Additional penalty coefficients are introduced to penalize queue overflow and excessive delay, respectively, thereby encouraging stable operation under stochastic arrivals. Together, these parameters define the relative importance of efficiency and delay-related objectives in the wireless scheduling task.

*Table 5.* System parameters for wireless user scheduling.

| PARAMETER | VALUE |
| --- | --- |
| MEAN ARRIVAL RATE ($\lambda_n$) | 0.045 |
| SHAPE PARAMETER FOR ARRIVAL ($k$) | 2.0 |
| CAPACITY SCALE ($\eta$) | 0.6 |
| THROUGHPUT WEIGHT ($\omega$) | 1.0 |
| OVERFLOW PENALTY COEFFICIENT ($\lambda_{\text{ov}}$) | 2.0 |
| DELAY PENALTY COEFFICIENT ($\lambda_{\text{delay}}$) | 1.0 |

**Inventory Replenishment System Parameters**   The inventory replenishment environment is governed by a set of parameters that control replenishment dynamics, demand uncertainty, and economic trade-offs. Specifically, the replenishment quantity determines the amount of inventory injected into the system at each decision step, while the demand granularity parameter controls the resolution of stochastic demand realizations. The demand and margin drift parameters introduce temporal variability in item characteristics, capturing non-stationary demand and pricing conditions. The ranges of unit cost, holding cost, and stock-out penalty define the item-specific economic factors that shape the trade-off between inventory availability and operational cost. An overflow penalty coefficient discourages excessive inventory accumulation beyond capacity. Finally, a set of reward weights balances revenue, ordering cost, holding cost, stock-out penalties, and overflow penalties. This defines the relative importance of competing economic objectives in the inventory replenishment task.

*Table 6.* System parameters for inventory replenishment.

| PARAMETER | VALUE |
| --- | --- |
| REPLENISHMENT QUANTITY ($Q$) | 0.5 |
| DEMAND GRANULARITY ($K_d$) | 50 |
| MEAN DEMAND DRIFT STD. ($\sigma_\mu$) | 0.02 |
| MARGIN DRIFT STD. ($\sigma_m$) | 0.003 |
| UNIT COST RANGE ($c_n$) | [0.02, 0.5] |
| HOLDING COST RANGE ($h_n$) | [0.01, 0.10] |
| STOCK-OUT PENALTY RANGE ($s_n$) | [0.05, 0.30] |
| OVERFLOW PENALTY COEFFICIENT ($\lambda$) | 1.0 |
| REWARD WEIGHTS $\{w_{\text{rev}}, w_{\text{ord}}, w_{\text{hold}}, w_{\text{so}}, w_{\text{ov}}\}$ | $\{2.0, 1.4, 3.0, 5.0, 3.5\}$ |

**Warehouse Clearance System Parameters**   The warehouse clearance environment is governed by a set of parameters that control global sales capacity, inventory inflow dynamics, and cost–revenue trade-offs. Specifically, the global sales capacity determines the total amount of inventory that can be cleared at each time step and scales linearly with the number of items, reflecting system-level resource constraints. Stochastic drift parameters for inflow rates and margins introduce temporal variability in inventory accumulation and economic value. A revenue scaling factor controls the contribution of realized sales to the overall reward, while a holding cost penalizes prolonged inventory accumulation. An overflow threshold defines the inventory level beyond which excessive stock is incurred, and an associated overflow penalty coefficient discourages persistent congestion. Together, these parameters define the balance between aggressive clearance for immediate revenue and conservative inventory management to avoid overflow penalties in the warehouse clearance task.

## E. Additional Results for Realistic Experiments

This appendix provides additional, detailed results for the realistic scheduling tasks, which support the realistic experiment in Section 5.2. In the performance results, the training system of size $N$ is denoted by $N^{\text{ID}}$, indicating which training system is used for training each policy. The FSP policy is trained only once by using the training system with $N = 10$, and therefore, there is only FSP($10^{\text{ID}}$). In contrast, the DQN policies are individually trained for each value of $N$ due to a

*Table 7.* System parameters for warehouse clearance.

| PARAMETER | VALUE |
|---|---|
| GLOBAL SALES CAPACITY ($C$) | $0.45 + 0.01 \cdot N$ |
| INFLOW RATE DRIFT STD. ($\sigma_\rho$) | 0.01 |
| MARGIN DRIFT STD. ($\sigma_m$) | 0.01 |
| REVENUE SCALING FACTOR ($\alpha$) | 5.0 |
| HOLDING COST ($h$) | 0.1 |
| OVERFLOW THRESHOLD ($\tau$) | 1.0 |
| OVERFLOW PENALTY COEFFICIENT ($\lambda$) | 5.0 |

lack of transferability. Thus, there are four DQN policies trained at different values of $N$, and each DQN policy cannot be applied to a system size different from the value of $N$ for which it was trained. In addition, to eliminate the disturbance from the uninformative random feature, we consider the DQN policy that excludes the random feature and uses only the three task-relevant state variables, which is denoted by DQN(w/o noise). The expanded comparison with Whittle, NAM, A2C, PPO, TRPO, and QR-DQN is summarized in the main text in Table 3. The appendix tables below focus on the detailed FSP–DQN ID/OOD breakdown from the original transfer analysis, including the DQN variant without the uninformative feature.

### E.1. Wireless User Scheduling

*Table 8.* In-distribution rewards and out-of-distribution average rewards of the wireless user scheduling task under varying $N$ with different system populations. (For each $N$, evaluation is conducted across 100 system instances.)

| $N$ | FSP($10^{\text{ID}}$) | DQN($5^{\text{ID}}$) | DQN($10^{\text{ID}}$) | DQN($15^{\text{ID}}$) | DQN($20^{\text{ID}}$) | DQN($10^{\text{ID}}$, W/O NOISE) |
|---|---|---|---|---|---|---|
| $5^{\text{ID}}$ | 0.116 | 0.127 | - | - | - | - |
| $10^{\text{ID}}$ | 0.109 | - | 0.112 | - | - | 0.122 |
| $15^{\text{ID}}$ | -0.266 | - | - | -0.297 | - | - |
| $20^{\text{ID}}$ | -0.762 | - | - | - | -0.855 | - |
| 5 | $0.085 \pm 0.001$ | $-0.073 \pm 0.002$ | - | - | - | - |
| 10 | $0.103 \pm 0.001$ | - | $-1.313 \pm 0.003$ | - | - | $-1.046 \pm 0.001$ |
| 15 | $-0.285 \pm 0.002$ | - | - | $-1.965 \pm 0.002$ | - | - |
| 20 | $-0.781 \pm 0.001$ | - | - | - | $-2.609 \pm 0.001$ | - |

Table 8 reports additional results for the wireless user scheduling task, providing a detailed comparison across different training system sizes and system populations. The upper block reports in-distribution (ID) rewards, where each DQN policy is evaluated on the training system with $N$ on which it is trained. The FSP policy is evaluated on the training systems with all values of $N$. The lower block reports out-of-distribution (OOD) evaluations under varying $N$. Note that each DQN policy is applicable only to the specific system size for which it is trained.

From the ID results, we observe that the DQN policies are not poorly trained: each DQN achieves strong performance on its corresponding training system. The FSP policy trained on the $10^{\text{ID}}$ system also achieves ID performance comparable to that of the DQN policies across different $N$'s, indicating that it does not sacrifice performance on the training system in exchange for transferability.

In contrast, the OOD results reveal a clear difference in generalization behavior. While the performance of DQN policies deteriorates substantially when evaluated on unseen system sizes or populations, the FSP policy maintains stable performance across all values of $N$. As a result, the FSP policy consistently outperforms the corresponding DQN policies in OOD settings, with the performance gap increasing as the system size grows.

We further provide the results of the DQN policy trained without the random noise feature. Although this variant achieves comparable ID performance to the standard DQN, it exhibits the same failure to generalize in OOD evaluations. This confirms that the lack of transferability in DQN is not due to noisy input features, but rather reflects a structural limitation of the policy representation.

Overall, these additional results reinforce the main finding that the scheduling principle learned by FSP is robust to changes in both system size and user population, whereas standard DQN policies remain tightly coupled to their training configurations.

Figures 7 and 8 provide the complete visualization of the learned additive and pairwise interaction components of the FSP policy for the wireless user scheduling task. The learned components exhibit structured patterns across the feature domain, implying that the FSP framework captures coherent prioritization rules.

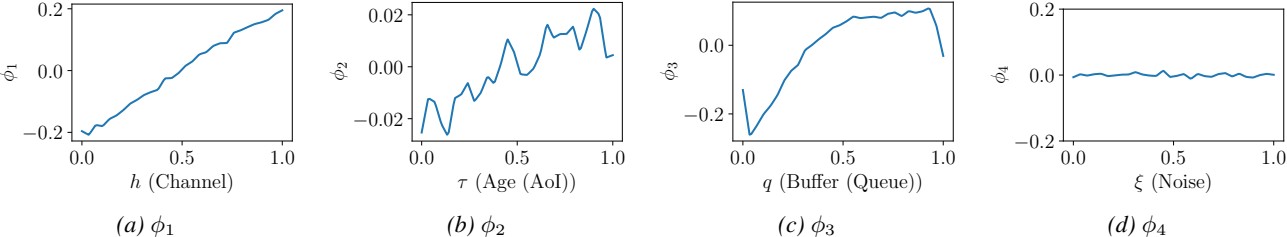

*(a) $\phi_1$*     *(b) $\phi_2$*     *(c) $\phi_3$*     *(d) $\phi_4$*

*Figure 7.* One-dimensional component functions of the FSP policy in the wireless user scheduling task.

Figure 7 summarizes the one-dimensional additive components associated with individual features. The components corresponding to informative system variables show clear and interpretable trends. As expected from the reward structure, in Figure 7a, $\phi_1(h)$ increases strongly and monotonically with the channel condition. In Figure 7b, $\phi_2(\tau)$, that corresponds to the age-of-information, increases, reflecting gradual changes in scheduling priority, but its effect is relatively small. Similarly, in Figure 7c, $\phi_3(q)$ increases smoothly with the buffer state, but its trend is not completely monotonic as in $\phi_1(h)$. In contrast, the component associated with the intentionally uninformative noise feature, $\phi_4(\xi)$, in Figure 7d remains nearly flat, confirming that the learned principle effectively suppresses irrelevant information.

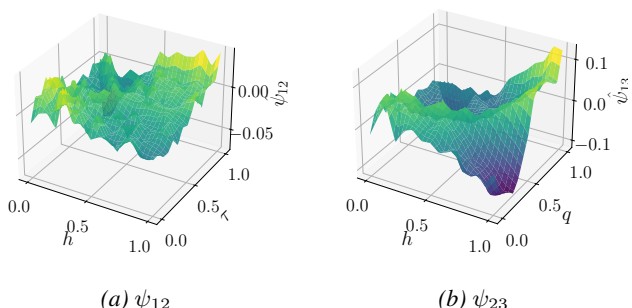

*(a) $\psi_{12}$*     *(b) $\psi_{23}$*

*Figure 8.* Two-dimensional pairwise interaction components of the FSP policy in the wireless user scheduling task.

Figure 8 visualizes the learned two-dimensional pairwise interaction components, which capture joint effects beyond purely additive contributions. The interaction surfaces are structured across the feature space, indicating consistent modulation of priorities based on combined feature states. In Figure 8a, while no sharply separated patterns are observed, the interaction reveals that users with large age-of-information and good channel conditions tend to receive higher priority. Figure 8b exhibits a clearer structure. Users with sufficiently occupied buffers and good channel conditions are assigned high priority, whereas users with poor channel conditions receive low priority even when their buffers are full. Moreover, when the buffer is nearly empty, the priority remains low, regardless of the channel condition, indicating that the policy avoids allocating resources to users with little backlog. Especially, the priority is significantly lower with good channel conditions. This behavior is consistent with the one-dimensional channel component in Figure 7a, where good channel quality already contributes positively through $\phi_1(h)$, and the interaction term further modulates this effect based on the buffer state.

Taken together, these visualizations provide a complete view of the learned scheduling principle. By jointly examining the one-dimensional additive components and the two-dimensional interaction terms, we can understand prioritization behaviors in the scheduling principle that cannot be captured by either component alone. This decomposition highlights how the FSP framework combines simple, interpretable effects into a coherent policy that captures more complex decision logic while maintaining stability and interpretability across the entire feature domain.

### E.2. Inventory Replenishment

Table 9 reports additional results for the inventory replenishment task, comparing the ID and OOD performance across different system sizes and system populations. The upper block reports ID rewards, where each DQN policy is evaluated on the training system with $N$ on which it is trained. The lower block reports OOD evaluations under varying $N$. Each DQN policy is applicable only to the specific system size for which it is trained.

*Table 9.* In-distribution rewards and out-of-distribution average rewards of the inventory replenishment task under varying $N$ with different system populations. (For each $N$, evaluation is conducted across 100 system instances.)

| $N$ | FSP($10^{\text{ID}}$) | DQN($5^{\text{ID}}$) | DQN($10^{\text{ID}}$) | DQN($15^{\text{ID}}$) | DQN($20^{\text{ID}}$) | DQN($10^{\text{ID}}$, W/O NOISE) |
|---|---|---|---|---|---|---|
| $5^{\text{ID}}$ | -0.232 | -0.255 | - | - | - | - |
| $10^{\text{ID}}$ | -0.353 | - | -0.355 | - | - | -0.363 |
| $15^{\text{ID}}$ | -0.649 | - | - | -0.670 | - | - |
| $20^{\text{ID}}$ | -0.769 | - | - | - | -0.751 | - |
| 5 | $-0.206 \pm 0.011$ | $-0.228 \pm 0.012$ | - | - | - | - |
| 10 | $-0.412 \pm 0.008$ | - | $-0.429 \pm 0.008$ | - | - | $-0.420 \pm 0.008$ |
| 15 | $-0.712 \pm 0.008$ | - | - | $-0.727 \pm 0.008$ | - | - |
| 20 | $-0.726 \pm 0.008$ | - | - | - | $-0.722 \pm 0.008$ | - |

From the ID results, both FSP and DQN policies achieve comparable performance on their respective training systems. This indicates that all policies are sufficiently trained for the inventory replenishment task. The FSP policy trained on the $10^{\text{ID}}$ system performs similarly to the DQN policies trained on their corresponding system sizes, regardless of the value of $N$. This shows the transferability of the FSP policy.

Unlike the wireless user scheduling task, the OOD results for inventory replenishment exhibit a different generalization pattern. For a fixed system size $N$, both FSP and DQN policies maintain similar performance when evaluated on different system populations. This suggests that the task admits a degree of transferability across population variations. This behavior is consistent with the quasi-additive structure of the inventory replenishment problem, where item-level decisions are weakly coupled. Across different values of $N$, the FSP policy maintains stable performance despite being trained only on the $10^{\text{ID}}$ system. In contrast, DQN policies generalize reasonably well across system populations for a fixed $N$. However, they remain tied to their respective training system sizes and cannot be transferred across different values of $N$.

We also present the DQN variant trained without the noise feature. Its performance is comparable to that of the standard DQN in both ID and OOD evaluations. This confirms that the observed generalization behavior is driven by the underlying task structure.

These results contrast sharply with those of the wireless user scheduling task. Although both FSP and DQN can generalize across system populations with a fixed system size in inventory replenishment, only FSP exhibits robustness to changes in the system size $N$, which highlights the benefit of learning a reusable scheduling principle.

Figures 9 and 10 provide the complete visualization of the learned additive and pairwise interaction components of the FSP policy for the inventory replenishment task. The learned components exhibit structured and economically meaningful patterns across the feature domain, indicating that the FSP framework captures coherent replenishment priorities aligned with the underlying cost and reward structure.

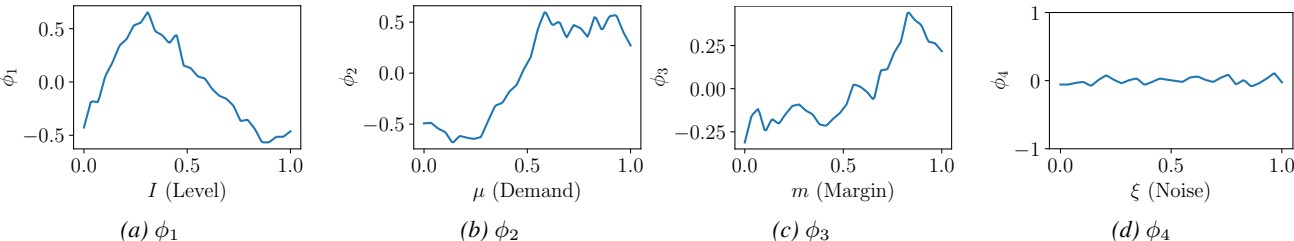

(a) $\phi_1$      (b) $\phi_2$      (c) $\phi_3$      (d) $\phi_4$

*Figure 9.* One-dimensional component functions of the FSP policy in the inventory replenishment environment.

Figure 9 summarizes the one-dimensional additive components associated with individual state variables. The relative

scale of the components suggests that demand-related information plays the dominant role in the learned scheduling principle, followed by inventory level and unit margin. Specifically, in Figure 9a, the inventory-level component $\phi_1$ exhibits an increasing-then-decreasing trend, indicating that items with low inventory are prioritized for replenishment, while items with excessive inventory are systematically deprioritized. The demand-related component $\phi_2$ in Figure 9b increases monotonically, reflecting a clear preference for items with higher expected demand. Similarly, the margin-related component $\phi_3$ in Figure 9b shows a positive correlation, indicating that higher-margin items receive higher priority when other factors are comparable. In contrast, Figure 9c shows that the component associated with the intentionally uninformative random feature, $\phi_4$, does not display a clear monotonic structure, confirming that the learned principle assigns negligible importance to irrelevant state dimensions.

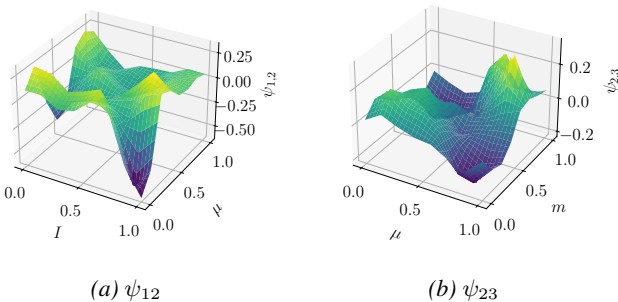

*(a) $\psi_{12}$*          *(b) $\psi_{23}$*

*Figure 10.* Learned two-dimensional interaction components $\psi_{ij}$ in the inventory replenishment environment, visualized as surfaces over the corresponding state dimensions.

Figure 10 visualizes the learned two-dimensional pairwise interaction components, which capture non-additive effects between state variables. The interaction surfaces exhibit smooth and structured patterns across the feature space, indicating consistent modulation of priorities based on joint state configurations. In Figure 10a, the interaction between inventory level and demand reveals that demand sensitivity is amplified when inventory is scarce, leading to strong prioritization of high-demand items to avoid stockouts. Conversely, when inventory is abundant and demand is low, the interaction becomes negative, discouraging further replenishment due to holding and overflow costs. Figure 10b shows that the importance of demand is further modulated by unit margin: high-demand items receive increasing priority as margin grows, while a mild positive interaction in low-demand, low-margin regions reflects a tendency to clear inventory to mitigate future holding risks.

Taken together, these visualizations offer an integrated view of the learned replenishment strategy. Considering the one-dimensional components alongside the pairwise interaction terms reveals how the policy jointly accounts for demand intensity, inventory dynamics, and revenue potential in its decision-making. This structured decomposition demonstrates that the FSP framework constructs a consistent and interpretable policy by combining simple effects in a way that captures richer economic trade-offs, while remaining stable across different systems.

### E.3. Warehouse Clearance

Table 10 reports additional transfer results for the warehouse clearance task under varying system sizes and system populations. The upper block of the table reports the ID rewards, where each policy is evaluated on the system size on which it is trained, while the lower block reports OOD evaluations across different values of $N$. As in previous experiments, each DQN policy is applicable only to the specific system size for which it is trained.

From the ID results, we see that both FSP and DQN policies achieve strong performance on their respective training systems. The DQN policies are not undertrained and slightly outperform the FSP policy. The FSP policy trained on the $10^{\text{ID}}$ system achieves competitive ID performance relative to the DQN policies. This confirms that it does not highly sacrifice ID optimality. However, the OOD results reveal a drastic difference in generalization behavior. When evaluated on unseen item populations, the performance of DQN policies deteriorates dramatically, often resulting in extremely large negative rewards. This degradation is particularly severe due to overflow penalties, which amplify the consequences of poor prioritization decisions when the policy fails to adapt to changes in feature distributions. In contrast, the FSP policy maintains stable and consistently positive performance across all values of $N$, despite being trained only on the $10^{\text{ID}}$ system. As a result, the FSP policy significantly outperforms the corresponding DQN policies in all OOD settings, demonstrating robust transferability across system sizes and populations.

*Table 10.* In-distribution rewards and out-of-distribution average rewards of the warehouse clearance task under varying $N$ with different system populations. (For each $N$, evaluation is conducted across 100 system instances.)

| $N$ | $\text{FSP}(10^{\text{ID}})$ | $\text{DQN}(5^{\text{ID}})$ | $\text{DQN}(10^{\text{ID}})$ | $\text{DQN}(15^{\text{ID}})$ | $\text{DQN}(20^{\text{ID}})$ | $\text{DQN}(10^{\text{ID}}, \text{w/o noise})$ |
|---|---|---|---|---|---|---|
| $5^{\text{ID}}$ | 0.736 | 0.772 | - | - | - | - |
| $10^{\text{ID}}$ | 0.823 | - | 0.883 | - | - | 0.881 |
| $15^{\text{ID}}$ | 0.739 | - | - | 0.808 | - | - |
| $20^{\text{ID}}$ | 0.723 | - | - | - | 0.766 | - |
| 5 | $0.630 \pm 0.021$ | $-10.913 \pm 1.374$ | - | - | - | - |
| 10 | $0.738 \pm 0.019$ | - | $-35.238 \pm 1.931$ | - | - | $-38.247 \pm 2.461$ |
| 15 | $0.712 \pm 0.014$ | - | - | $-16.747 \pm 2.807$ | - | - |
| 20 | $0.649 \pm 0.023$ | - | - | - | $-9.063 \pm 1.913$ | - |

As in the previous environments, we provide a DQN variant trained without the noise feature. While its ID performance remains comparable to that of the standard DQN, it exhibits the same catastrophic performance degradation in OOD evaluations. This confirms that the failure of DQN to generalize in this task is not caused by noisy input features, but rather reflects a structural limitation in handling overflow-sensitive dynamics.

At first glance, this task seems similar to the inventory replenishment task, but these additional results reinforce the findings observed in the wireless user scheduling task. When the task involves strong nonlinear penalties and tight coupling between system size and decision quality, transferable scheduling principles learned by the FSP framework provide substantial robustness advantages over standard DQN policies.

Figures 11 and 12 provide the complete visualization of the learned additive and pairwise interaction components of the FSP policy for the warehouse clearance task.

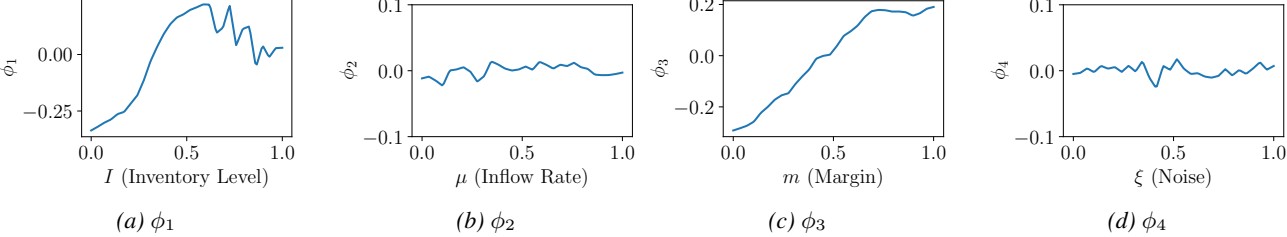

(a) $\phi_1$        (b) $\phi_2$        (c) $\phi_3$        (d) $\phi_4$

*Figure 11.* One-dimensional component functions of the FSP policy in the warehouse clearance environment.

Figure 11 summarizes the one-dimensional additive components associated with individual state variables. From Figure 11a, we can see that the inventory-level component $\phi_1(I)$ increases for small to moderate inventory levels and then slightly decreases as the inventory approaches the overflow threshold. This non-monotonic shape indicates the non-purely additive effect of the inventory level and the presence of interaction effects. Figure 11b corresponds to the inflow rate. The inflow rate component $\phi_2(\mu)$ has a small magnitude and lacks a clear monotonic trend. This suggests that inflow information plays a limited independent role. This behavior implies that the current inventory level provides a more immediate and dominant signal of overflow risk compared with the predictive role of inflow at the decision time. In contrast, the margin-related component $\phi_3(m)$ shown in Figure 11c increases smoothly with margin, reflecting a consistent preference for higher-margin items when other state variables are comparable. Finally, in Figure 11d, the component associated with the intentionally uninformative noise feature, $\phi_4(\xi)$, remains nearly flat, confirming that the learned principle suppresses irrelevant information.

Figure 12 visualizes the learned two-dimensional pairwise interaction components that capture joint effects beyond purely additive contributions. The interaction surfaces are structured across the feature space, indicating systematic modulation of priorities based on combined state configurations. Figures 12a and 12c show that interactions involving the inflow rate have weak structure, which is consistent with the limited role of inflow observed in the one-dimensional components. In contrast, the interaction between inventory level and margin, shown in Figure 12b, exhibits an interpretable pattern that complements the non-monotonic behavior of the inventory-level component $\phi_1(I)$. For moderate inventory levels, items with higher margins are given significantly higher priority, whereas items with low margins are deprioritized. While the one-dimensional

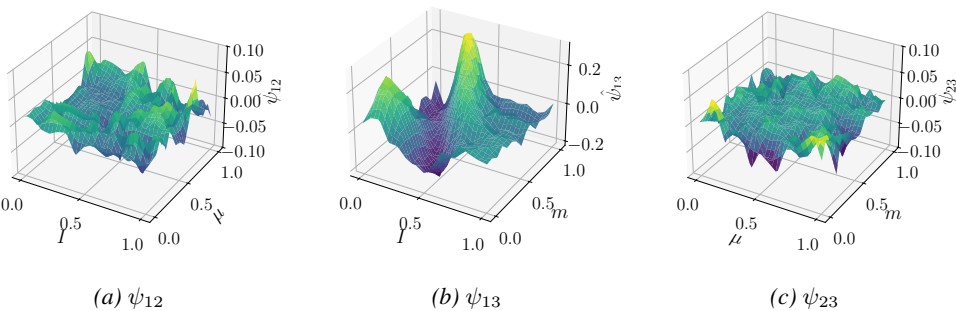

*(a) $\psi_{12}$*        *(b) $\psi_{13}$*        *(c) $\psi_{23}$*

*Figure 12.* Learned two-dimensional interaction components $\psi_{ij}$ in the warehouse clearance environment, visualized as surfaces over the corresponding state dimensions.

component $\phi_1(I)$ alone appears to have a higher priority around moderate inventory levels, the interaction reveals that this preference is conditional on margin. When inventory levels are low, even high-margin items are assigned low priorities. This implies that the policy avoids allocating clearance capacity to items that do not generate sufficient revenue due to low inventory levels. Together with the shape of $\phi_1(I)$, this interaction indicates that the influence of inventory levels is not purely additive, but is selectively amplified or suppressed depending on margin.

These visualizations provide an integrated view of the learned clearance strategy. Examining the one-dimensional additive components together with the pairwise interaction terms clarifies how the policy coordinates inventory pressure and revenue signals in its prioritization decisions. This decomposition illustrates that the FSP framework constructs a coherent and interpretable policy by combining simple effects to represent more complex decision logic, while maintaining transferability.

