# OpenReview forum: "Factorized Scheduling Principle: Learning Interpretable and Transferable Policies via Structured Additive Functions"
_ICML.cc/2026/Conference — ICML 2026 regular_

### Official Review · Reviewer_koej · 2026-02-16

**Soundness:** 2
**Presentation:** 2
**Significance:** 2
**Originality:** 3
**Overall Recommendation:** 4
**Confidence:** 1

**Summary:**

This paper proposes the Factorized Scheduling Principle (FSP), which learns an interpretable global priority function instead of a black-box RL policy. The authors intend to analyze a central aspect of scheduling: whether reusable priority rules can be learned directly from experience. The work strives to present a major issue in current RL schedulers, namely poor interpretability and transferability. FSP models priorities as a structured additive function with identifiability constraints and trains it using likelihood and temporal-difference losses. Theoretical guarantees and experiments support interpretability and zero-shot transfer across system sizes.

**Compliance With Llm Reviewing Policy:**

Affirmed.

**Final Justification:**

I am not the expert in the paper related topic. The questions of my review have been answered.

**Key Questions For Authors:**

Can it extend to multi-item selection settings?

**Limitations:**

yes

**Strengths And Weaknesses:**

S

The idea of learning an explicit scheduling principle rather than a policy is clear and well-motivated. The additive structure with centering constraints provides strong interpretability. The theoretical analysis is clean and covers identifiability and ranking consistency. Experiments demonstrate strong transfer across different candidate sizes. Overall, the framework is conceptually coherent and addresses an important gap between classical index rules and RL methods.

W

Since I am not an expert in this field, I may misunderstand some details, but the assumption of a global additive principle seems strong for complex coupled systems. The experimental comparison is mainly against DQN, and stronger permutation-invariant or structured baselines would strengthen the claims. Scalability with higher-dimensional features and many interaction terms is also unclear.

---

> ### Author Rebuttal · Authors · 2026-03-31
>
> We thank the reviewer for the assessment and constructive feedback.
>
> ---
>
> ### Additive assumption in complex systems
>
> We agree that assuming a strictly additive global principle may appear strong for complex coupled systems. However, we emphasize that in item-selection scheduling problems, there are structural reasons why an additive principle can serve as a meaningful approximation. In many such systems:
>
> - rewards are primarily determined by the selected item’s condition,
> - interactions with other items often appear indirectly through constraints or aggregated effects, and
> - decisions are ultimately driven by relative ranking among items.
>
> This structure naturally aligns with representing priorities through an item-level scoring function $S(x)$, as also reflected in classical index-based approaches. Therefore, the additive structure should be understood as a structured and interpretable approximation aligned with the problem structure, rather than a strict assumption about the environment.
>
> Furthermore, our realistic experiments demonstrate that FSP remains effective even when this structure is violated: the wireless scheduling task involves strong global coupling, and the warehouse clearance task exhibits highly non-additive dynamics. Despite this, FSP maintains stable performance and transferability, indicating that it captures dominant decision structure beyond strict additivity.
>
> ### Baselines
>
> We appreciate the suggestion to include permutation-invariant or structured baselines. FSP is not a specific function class but a framework that defines a new policy structure via a global principle $S(x)$. Methods such as neural additive models or permutation-invariant architectures are orthogonal to our approach and can be incorporated within the FSP framework rather than serving as direct alternatives. Therefore, our comparison with DQN is intended to highlight the contrast between standard policy-based RL (entangled, non-transferable), and principle-based scheduling (structured, transferable).
>
> In this context, Whittle index produces per-item priority scores as in our method. However, it does not inherently guarantee transferability across varying $N$ and relies on fundamentally different assumptions such as the independent arm dynamics and model knowledge. These imply that Whittle index is not directly applicable without modification in our setting. So here we designed a Whittle index-based baseline, which can reuse the Whittle index table learned at a fixed system size for different $N$, and provide its results for the inventory task for completeness. Due to time constraints during the rebuttal period, results on additional tasks are still being finalized and will be included in the revision. (Whittle refers to a policy trained at $N=10$ and directly transferring it to other problem sizes without re-computation.)
>
> | N | FSP | DQN | Whittle | Random |
> | --- | --- | --- | --- | --- |
> | 5 | -0.206 ± 0.011 | -0.228 ± 0.012 | -0.231 ± 0.010 | -0.307 ± 0.013 |
> | 10 | -0.412 ± 0.008 | -0.429 ± 0.008 | -0.405 ± 0.009 | -0.505 ± 0.009 |
> | 15 | -0.712 ± 0.008 | -0.727 ± 0.008 | -0.723 ± 0.008 | -0.801 ± 0.008 |
> | 20 | -0.726 ± 0.008 | -0.742 ± 0.008 | -0.921 ± 0.011 | -0.812 ± 0.008 |
>
> The results show that in most settings FSP matches or outperforms the transfer-based Whittle baseline. Notably, the performance of the Whittle baseline shows noticeable degradation as $N$ increases (e.g., $N=20$), indicating that index policies estimated under a fixed system configuration may not generalize well across different candidate set sizes. In contrast, FSP maintains stable performance across all $N$, which suggests that learning a feature-level scheduling principle enables more robust transfer across varying system sizes without relying on fixed index structures. In addition, FSP provides an interpretable decomposition of the decision rule, which is not available in index-based methods.
>
> ### Scalability
>
> We agree that scalability with respect to feature dimension and interaction complexity is important. FSP is designed to helps alleviate the curse of dimensionality by decomposing the principle into univariate and pairwise components, resulting in $O(K + |P|)$ complexity. Moreover, the interaction set $P$ can be restricted to a sparse subset. While current experiments use $K=4$, we demonstrate strong scalability with respect to the number of items $N$, which is the main combinatorial challenge in scheduling. Extending to higher-dimensional feature spaces via sparse interactions or low-rank structures is a promising direction.
>
> ### Extension to multi-item selection
>
> We thank the reviewer for this insightful question. As we discussed in Section 6, FSP can be extended to multi-item selection settings. In such cases, the learned principle $S(x)$ can be used as a reusable scoring function, which can be combined with standard combinatorial selection methods (e.g., top-k selection, knapsack-style constraints).

---

> > ### Author Rebuttal · Reviewer_koej · 2026-04-02
> >
> > Thanks for the explanation. I do not have further questions.

---

> > > ### Author Response · Authors · 2026-04-07
> > >
> > > Thank you for your careful review and for updating your assessment. We sincerely appreciate your time and feedback.

---

### Official Review · Reviewer_QFsH · 2026-03-08

**Soundness:** 2
**Presentation:** 3
**Significance:** 2
**Originality:** 2
**Overall Recommendation:** 3
**Confidence:** 4

**Summary:**

This paper proposes the FSP framework, which aims to learn an interpretable and transferable global priority rule for scheduling problems, where a system must repeatedly select one item from a set of candidates. The key idea is to learn a function, termed the global scheduling principle S(x), defined directly on the feature vector of a single item, rather than a policy dependent on the concatenated state of all items.

This principle is structured as an additive sum of univariate and bivariate smooth functions, ensuring identifiability through centering constraints. The framework is trained using a combination of policy likelihood and temporal-difference value losses. Theoretical guarantees regarding identifiability, approximation error, and transfer invariance are provided. Experiments on tasks demonstrate that the learned principle can be applied zero-shot to systems with different numbers of items, maintaining performance while offering interpretability of decisions through its decomposed feature contributions.

**Compliance With Llm Reviewing Policy:**

Affirmed.

**Final Justification:**

As noted in my comments, the reviewers' response does not address concerns regarding the generalizability of item-selection scheduling to other problems, such as packing. Furthermore, the unique design of this feature has not been introduced in other similar methods to demonstrate its effectiveness.

**Key Questions For Authors:**

Item-level feature design is akin to constructing independent features for each action. While this can obviously solve the scalability issue caused by an increasing number of actions, it also artificially separates and loses the interrelationships between actions. This seems to be more disadvantageous in sequential decision-making processes.

**Limitations:**

The analysis of the limitations of FSP is insufficient. For example, the impact of the loss of features between actions was not discussed, and the computation time was not further analyzed.

**Strengths And Weaknesses:**

Strengths

The paper reflects on existing state-reaction-based methods in terms of transferability and interpretability, exploring a priority principle constructed around task features, which brings certain improvements.
The state definition of this work is relatively clear, and it also provides some theoretical analysis of FSP.

Weaknesses

FSP and the problems it intends to solve are not broadly applicable to various scheduling problems, but only to a limited subset. Therefore, it is recommended to adjust the description or supplement proofs regarding generalization.
The authors' review of existing reinforcement learning methods is not sufficiently clear or deep. Clearly, different MDP modeling approaches lead to diverse definitions of the state space, not merely simple concatenation of item information. Examples include observation features designed from different perspectives of the problem, feature representations based on disjunctive graphs, etc.
FSP involves several key parameters, and further sensitivity experiments and analysis should be conducted on them.
The scenarios addressed in this work are simple, which does little to alleviate readers' concerns about the method's extensibility.

---

> ### Author Rebuttal · Authors · 2026-03-31
>
> We thank the reviewer for the thoughtful feedback and clarify the points below.
>
> ---
>
> ### Applicability and generality
>
> We note that it is standard in the literature that different methods target different problem classes. For example, reinforcement learning methods are typically designed for general MDPs, but in practice their performance and applicability depend heavily on state representation, action structure, and training distribution. This has led to a wide range of RL variants tailored to different problem settings.
>
> In contrast, FSP is designed for a specific but broadly relevant class of scheduling problems. This class includes many practical applications (e.g., wireless scheduling, inventory control, recommendation, manufacturing systems, smart grids, etc.). Specifically, our work explicitly focuses on item-selection scheduling problems, and proposes a formulation that directly captures their underlying structure through a global priority principle $S(x)$. Therefore, the scope of FSP is not a limitation but a deliberate design choice. Rather than addressing all possible sequential decision problems, we aim to provide a structured and transferable solution for a well-defined and practically important subclass of applications.
>
> ### State representation and related RL formulations
>
> We agree that there exist various state representations in RL beyond simple concatenation. However, our point is not about the specific choice of encoding, but about the structure of the learned decision rule. Regardless of the representation (concatenation, graphs, or other structured features), standard RL approaches typically learn policies or value functions defined over the joint state, where decision logic is entangled across items. As a result, the contribution of individual item features cannot be isolated into a shared, reusable priority rule, and the learned policy remains tightly coupled to the candidate set configuration, limiting transferability.
>
> In contrast, our approach explicitly models a global item-level principle $S(x)$ that assigns scores independently of item identity or set size, enabling both interpretability and transfer across different candidate sets. Our contribution lies in decoupling decision logic at the item level, rather than proposing a new encoding scheme.
>
> ### Action interdependence and item-level abstraction
>
> We would like to clarify that action interdependence is not ignored in our framework. Our work explicitly focuses on a well-defined class of problems, item-selection-based scheduling. In this setting, dependencies between actions primarily arise from how items are compared within a candidate set, i.e., through their relative priority. FSP captures this by learning a global principle $S(x)$ that induces a consistent ranking across items.
>
> In addition, there exist dependencies arising from system dynamics and long-term effects. These are explicitly incorporated through the value component $V(\bar{s})$, which captures temporal consistency across states. Specifically, in the loss function, the likelihood term shapes local ranking among items, while the value term enforces global consistency of decisions over time. Therefore, our approach structures action dependencies through ranking and temporal consistency, while learning a transferable priority principle.
>
> ### Experimental scope and extensibility
>
> We respectfully note that the purpose of our experiments is not to cover all scheduling variants, but to demonstrate recoverability of the underlying principle (synthetic), interpretability of learned components, and transferability across system sizes and populations (realistic tasks). Moreover, in scheduling problems, complexity does not arise solely from the number of features, but from interaction structure and system dynamics. The considered scenarios are not purely simplistic. For example, the wireless scheduling task exhibits strong nonlinearity and global coupling, while the inventory task represents a quasi-additive regime. We agree that extending experiments to broader settings would be valuable, but we believe the current experiments are sufficient to demonstrate the key properties of the proposed framework.
>
> ### Sensitivity to parameters
>
> We agree that sensitivity analysis is important. As we answered in the rebuttal of Reviewer LALu, the hyperparameters of FSP are not that sensitive, and we will add more analysis on that in the revised version of the manuscript.
>
> ### Computation time
>
> In our implementation, the average computation time per training step is FSP≈0.0029 sec and DQN≈0.0026 sec, showing that the two methods have comparable computational cost in practice. While our current implementation is a straightforward prototype, the computational cost is already comparable, and can be further improved using standard optimization techniques.

---

> > ### Author Rebuttal · Reviewer_QFsH · 2026-04-02
> >
> > I have read the author’s rebuttal, but some concerns cannot be adequately addressed by explanations alone.
> >
> > The authors have helpfully added scalability tests for varying N, which is good. But the generalizability of the method to broader problem scenarios and comparisons with SOTA approaches remain lacking in statistical validation.
> >
> > In the rebuttal, the authors note that the method “aims to provide a structured and transferable solution for a well‑defined and practically important subclass of applications,” and in the manuscript this is a methodology centered around general problems and MDP design—both of which suggest the method could be effective in similar settings.
> >
> > To address potential concerns of over‑fitting and to better demonstrate real‑world applicability, it would be valuable to conduct additional experiments on another representative problem, such as bin packing.
> > Moreover, comparisons and discussion with the latest DRL methods are insufficient. DQN is a commonly used baseline, but there are many more advanced policy‑based and value‑based DRL approaches that should be included for a more comprehensive evaluation.

---

> > > ### Author Response · Authors · 2026-04-07
> > >
> > > ### Scope of FSP
> > >
> > > We would like to clarify that our work does not aim to address general MDPs, but focuses on a specific and well-defined class of problems: item-selection scheduling, where each action corresponds to selecting an item from a candidate set based on item-level conditions. This class includes many practical systems (e.g., wireless scheduling, inventory control, and recommendation), and the selected tasks are representative of this problem class and sufficient to demonstrate the key properties of the proposed framework.
> > >
> > > ### Additional experiments
> > >
> > > While many advanced DRL methods exist, they largely follow the same paradigm as DQN, and do not explicitly address transferability across varying candidate sets or feature-level interpretability. Thus, we initially compare against DQN as a representative baseline. However, we agree that the comparison only with DQN is insufficient to clearly show the impact of the FSP framework.
> > >
> > > As the reviewer suggested, we further include a range of SOTA standard DRL baselines (A2C, PPO, TRPO, QR-DQN) across tasks, along with additional structured baselines to provide a more comprehensive empirical comparison: The NAM-based DQN uses a feature-additive Q-network, providing a more interpretable structure compared to standard DQN. The Whittle index adopts a per-item scoring rule, enabling limited transfer across varying N. A qualitative comparison is summarized below.
> > >
> > > | Baseline | Transferability | Interpretability |
> > > | --- | --- | --- |
> > > | Standard DRL | ❌ | ❌ |
> > > | NAM | ❌ | ✅ |
> > > | Whittle | ✅ | ❌ |
> > > | FSP (ours) | ✅ | ✅ |
> > >
> > > We emphasize that the comparison setting highlights differences in transferability: FSP and Whittle are trained once at $N=10$ and evaluated across varying both system sizes and dynamics without retraining, whereas others are retrained for each $N$ and only generalize over system dynamics. Due to limited time during the rebuttal period, SOTA DRL baselines are not yet included for the inventory task, but will be added them in revision. Nevertheless, the current results already provide a representative comparison.
> > >
> > > **Wireless user scheduling**
> > >
> > > | N | FSP | Whittle | NAM | DQN | A2C	 | PPO	 | TRPO	 | QR-DQN	 |
> > > | --- | --- | --- | --- | --- | --- | --- | --- | --- |
> > > | 5 | 0.085 ± 0.001 | 0.096 ± 0.000 | 0.131 ± 0.000 | -0.073 ± 0.002 | 0.127 ± 0.000 | 0.129 ± 0.000	 | 0.130 ± 0.000	 | 0.126 ± 0.000	 |
> > > | 10 | 0.103 ± 0.001 | 0.115 ± 0.001 | 0.109 ± 0.001 | -1.313 ± 0.003 | 0.095 ± 0.001 | 0.104 ± 0.001	 | 0.112 ± 0.001	 | 0.119 ± 0.001	 |
> > > | 15 | -0.285 ± 0.002 | -0.337 ± 0.002 | -0.251 ± 0.001 | -1.965 ± 0.002 | -0.426 ± 0.002 | -0.372 ± 0.002 | -0.368 ± 0.002	 | -0.386 ± 0.002	 |
> > > | 20 | -0.781 ± 0.001 | -0.888 ± 0.001 | -0.752 ± 0.001 | -2.609 ± 0.001 | -1.045 ± 0.002 | -0.971 ± 0.002	 | -0.951 ± 0.002 | -1.019 ± 0.002	 |
> > >
> > > **Warehouse clearance**
> > >
> > > | N | FSP | Whittle | NAM | DQN | A2C | PPO | TRPO | QR-DQN |
> > > | --- | --- | --- | --- | --- | --- | --- | --- | --- |
> > > | 5 | 0.630 ± 0.021 | 0.596 ± 0.021 | -9.272 ± 0.920 | -10.913 ± 1.374 | -1.677 ± 0.714 | -9.587 ± 1.041 | -4.789 ± 1.086 | -0.610 ± 0.289 |
> > > | 10 | 0.738 ± 0.019 | 0.677 ± 0.019 | -18.031 ± 1.269 | -35.238 ± 1.931 | 0.682 ± 0.021 | -7.817 ± 1.470 | 0.696 ± 0.021 | -1.974 ± 0.457 |
> > > | 15 | 0.712 ± 0.014 | 0.621 ± 0.014 | -25.995 ± 2.208 | -16.747 ± 2.807 | -68.798 ± 0.030 | -0.931 ± 0.375 | 0.601 ± 0.015 | 0.637 ± 0.015 |
> > > | 20 | 0.649 ± 0.023 | 0.605 ± 0.015 | -30.154 ± 2.545 | -9.063 ± 1.913 | -92.641 ± 0.035 | -83.264 ± 0.118 | 0.545 ± 0.015 | -63.028 ± 0.412 |
> > >
> > > **Inventory replenishment**
> > >
> > > | N | FSP | Whittle | NAM | DQN |
> > > | --- | --- | --- | --- | --- |
> > > | 5 | -0.206 ± 0.011 | -0.231 ± 0.010 | -0.232 ± 0.014 | -0.228 ± 0.012 |
> > > | 10 | -0.412 ± 0.008 | -0.405 ± 0.009 | -0.456 ± 0.018 | -0.429 ± 0.008 |
> > > | 15 | -0.712 ± 0.008 | -0.723 ± 0.008 | -0.763 ± 0.017 | -0.727 ± 0.008 |
> > > | 20 | -0.726 ± 0.008 | -0.921 ± 0.011 | -0.849 ± 0.022 | -0.742 ± 0.008 |
> > >
> > > Despite this easier setting, DRL baselines and NAM often show limited generalization similar to DQN, which suggests that these methods do not explicitly incorporate structural principle tailored to item-selection scheduling problems. On the other hand, Whittle exhibits partial transferability and can be competitive in small-scale settings (e.g., $N=5,10$), but its performance degrades as $N$ increases, falling behind FSP in larger or more coupled environments. We also note that FSP provides an additional advantage over Whittle beyond performance: interpretability. FSP yields an explicit factorized representation in terms of item-level and interaction-level components, making it possible to interpret how the policy prioritizes items and captures dependencies. These additional results consistently show that FSP maintains stable performance across varying system sizes, while also providing interpretability, whereas standard methods exhibit degradation.

---

### Official Review · Reviewer_LALu · 2026-03-13

**Soundness:** 3
**Presentation:** 3
**Significance:** 3
**Originality:** 3
**Overall Recommendation:** 4
**Confidence:** 3

**Summary:**

This paper proposes FSP, a framework for learning interpretable and transferable scheduling rules. It decomposes a global principle $S(x)$ into additive univariate $\phi_k(x_k)$ and pairwise $\psi_{kl}(x_k, x_l)$ with centering constraints. State is a condition distribution over item features, invariant to set size and ordering. Theory covers identifiability (Theorem 3.1), approximation (3.2), ranking consistency (3.3), and transfer (3.4). Experiments on synthetic and two realistic tasks show strong performance, interpretability, and zero-shot generalization across $N$.

**Compliance With Llm Reviewing Policy:**

Affirmed.

**Key Questions For Authors:**

See above weakness part.

**Limitations:**

Section 6 acknowledges the global principle assumption and notes FSP learns an approximate surrogate when exact additive structure is absent. But the gap between theoretical assumptions (A1, A2) and experimental settings is not discussed. Neither is the limited baseline comparison or scalability to $K \gg 4$.

**Strengths And Weaknesses:**

Strengths:

Fills a real gap between index-based scheduling (interpretable, rigid) and RL-based (flexible, opaque). Table 1 positions FSP clearly.

Additive decomposition with centering works well. Theorem 3.1 ensures unique components. Theorems 3.2, 3.3 connect approximation error to regret.

Transferability stands out. Wireless scheduling (Table 3): FSP trained on $N=10$ beats DQN trained per $N$, gaps +0.158 to +1.828. Warehouse clearance: DQN collapses to $-35.2$, FSP stays positive.

Visualizations (Figures 3, 7-12) are useful. Noise feature $\xi$ gets near-zero weight. Pairwise surfaces show non-trivial logic.

Weaknesses:

Assumption (A1) requires a ground-truth additive principle $S^\star$. The realistic experiments show FSP works when the environment is quasi-additive or even coupled, but no analysis of how performance degrades as the reward structure deviates from additivity. A controlled experiment varying non-additivity degree would help.

Assumption (A2) requires weak interaction among items. The wireless task violates this (AoI penalty couples all users), yet FSP works well. This gap between theory and experiments is not discussed. The guarantees may not apply where FSP performs best.

Only DQN as baseline. No comparison with Whittle index methods (Fu et al., 2019; Robledo Relaño et al., 2024; Xiong & Li, 2023) or neural additive models for RL (Siems et al., 2023). These are the most natural competitors.

No sensitivity analysis for $\lambda$ (trade-off between $L_{\rm lik}$ and $L_{\rm val}$). Paper uses $\lambda = 0.1$ (Table 4) without justification.

The kernel $\kappa$ for the condition distribution is never specified. What was used? How sensitive are results to this choice?

---

> ### Author Rebuttal · Authors · 2026-03-31
>
> We thank the reviewer for the positive assessment and insightful feedback. We address the concerns below and will clarify these in the revision.
>
> ---
>
> ### Additivity assumption (A1) and robustness
>
> We agree that a controlled experiment would provide additional insight. However, we respectfully note that our current experiments already provide this evidence in a practically meaningful way. Our realistic experiments already span a spectrum of non-additivity, from quasi-additive with relatively weak interactions (inventory) to strongly coupled (wireless) and highly non-additive thresholded dynamics (warehouse clearance). We believe that such task-level variation provides more realistic and policy-relevant validation than a synthetic interpolation of interaction strength, since real scheduling systems often involve structured and heterogeneous forms of non-additivity rather than a single tunable parameter.
>
> ### Weak interaction assumption (A2)
>
> We first clarify that Assumption (A2) is not required for identifiability of the additive decomposition, but is introduced to ensure that a global scheduling principle $S^{\*}(x)$ is well-defined, i.e., that the reward can be consistently attributed to individual item features independent of other items. In particular, A2 serves as a sufficient condition to ensure the idealized, clean reduction to $S^{\*}(x)$. Therefore, in practice, our method does not rely on A2. In such cases, FSP instead learns an approximate surrogate principle that induces stable and transferable ranking across items. This is consistent with our empirical findings, where FSP maintains strong performance and transferability even in environments with pronounced interactions.
>
> ### Baselines
>
> We appreciate the reviewer’s suggestion regarding additional baselines such as neural additive models (NAMs) and Whittle index. First, we would like to clarify that FSP operates at a different level from NAMs. FSP is not a specific function approximator, but a framework that redefines the problem formulation, policy structure, and learning objective by introducing a global scheduling principle $S(x)$. In contrast, NAM is a function class that can be used to parameterize a mapping (e.g., a value function or policy), but does not by itself define how scheduling decisions are structured or how transfer across candidate sets is achieved. On the other hand, Whittle index produces per-item priority scores as in our method. However, it does not inherently guarantee transferability across varying N and relies on fundamentally different assumptions such as the independent arm dynamics and model knowledge. These imply that Whittle index is not directly applicable without modification in our setting. Nevertheless, we agree with the reviewer that Whittle index can be considered as a per-item scoring baseline. So here we designed a Whittle index-based baseline, which can reuse the Whittle index table learned at a fixed system size for different N, and provide its results for the inventory task for completeness. Due to space constraints, we refer the detailed results to our response to Reviewer koej.
>
> ### Hyperparameter choice and sensitivity
>
> First, regarding the trade-off parameter λ, if λ is too small, temporal consistency is underutilized. If λ is too large, it may weaken the sharpness of local ranking. In FSP learning, temporal consistency primarily stabilizes learning, rather than directly determining the ranking. Thus, we selected an appropriate value of λ to stabilize learning. Policy performance remains largely unchanged over a moderate range around the chosen value.
>
> Regarding the kernel formulation, we apologize for the lack of clarity. The kernel representation in the paper is introduced as a conceptual smoothing perspective that facilitates continuous interpretation and theoretical analysis. In our implementation, however, we operate directly on the empirical distribution of item features, i.e., a sum of Dirac masses. This empirical formulation is consistent with the integral form under the empirical measure and already provides a permutation-invariant representation of the candidate set. Importantly, this implementation is also computationally efficient, as it avoids evaluating kernel functions and reduces the computation to simple summations over items. As a result, the method does not depend on a specific kernel choice, and its performance is not sensitive to this aspect.
>
> ### Scalability
>
> FSP is designed to helps alleviate the curse of dimensionality by decomposing the principle into univariate and pairwise components, resulting in $O(K + |P|)$ complexity. Moreover, the interaction set $P$ can be restricted to a sparse subset. While current experiments use $K=4$, we demonstrate strong scalability with respect to the number of items N, which is the main combinatorial challenge in scheduling. Extending to higher-dimensional feature spaces via sparse interactions or low-rank structures is a promising direction.

---

> > ### Author Rebuttal · Reviewer_LALu · 2026-04-03
> >
> > Thank the author for the response. Can the authors provide Whittle index results on the wireless and warehouse clearance tasks, where the transferability gap is larger? DQN remains the only baseline in the main paper.

---

> > > ### Author Response · Authors · 2026-04-07
> > >
> > > Thank you for the helpful follow-up. As suggested, we additionally evaluated a Whittle-based baseline on the wireless scheduling and warehouse clearance tasks, where transferability gaps are more pronounced.
> > >
> > > **Wireless user scheduling**
> > >
> > > | N | FSP | Whittle | DQN |
> > > | --- | --- | --- | --- |
> > > | 5 | 0.085 ± 0.001 | 0.096 ± 0.000 | -0.073 ± 0.002 |
> > > | 10 | 0.103 ± 0.001 | 0.115 ± 0.001 | -1.313 ± 0.003 |
> > > | 15 | -0.285 ± 0.002 | -0.337 ± 0.002 | -1.965 ± 0.002 |
> > > | 20 | -0.781 ± 0.001 | -0.888 ± 0.001 | -2.609 ± 0.001 |
> > >
> > > **Inventory replenishment**
> > >
> > > | N | FSP | Whittle | DQN |
> > > | --- | --- | --- | --- |
> > > | 5 | -0.206 ± 0.011 | -0.231 ± 0.010 | -0.228 ± 0.012 |
> > > | 10 | -0.412 ± 0.008 | -0.405 ± 0.009 | -0.429 ± 0.008 |
> > > | 15 | -0.712 ± 0.008 | -0.723 ± 0.008 | -0.727 ± 0.008 |
> > > | 20 | -0.726 ± 0.008 | -0.921 ± 0.011 | -0.742 ± 0.008 |
> > >
> > > **Warehouse clearance**
> > >
> > > | N | FSP | Whittle | DQN |
> > > | --- | --- | --- | --- |
> > > | 5 | 0.630 ± 0.021 | 0.596 ± 0.021 | -10.913 ± 1.374 |
> > > | 10 | 0.738 ± 0.019 | 0.677 ± 0.019 | -35.238 ± 1.931 |
> > > | 15 | 0.712 ± 0.014 | 0.621 ± 0.014 | -16.747 ± 2.807 |
> > > | 20 | 0.649 ± 0.023 | 0.605 ± 0.015 | -9.063 ± 1.913 |
> > >
> > > As shown in the previous rebuttal with the inventory task, Whittle exhibits partial transferability and can be competitive in small-scale settings (e.g., $N=5,10$) in general, but its performance degrades as $N$ increases, falling behind FSP in larger or more coupled environments. This suggests that index policies derived under a fixed system configuration may not generalize robustly across varying $N$, particularly in more coupled environments. In contrast, FSP learns a global scheduling principle on item features, capturing both marginal effects and interactions. This results in more robust and scalable prioritization, achieving both transferability and improved performance.
> > >
> > > For completeness, we note that additional experiments with other structured and DRL baselines are provided in our response to Reviewer QFsH.
> > >
> > > We will include these additional results and discussion in the revision.

---

### Decision · Program_Chairs · 2026-04-30

**Decision:**

Accept (regular)

**Comment:**

This paper introduces a Factorized Scheduling Principle (FSP) framework for learning interpretable and transferable rules for scheduling problems, in settings where a system must repeatedly select one item from a set of candidates. The proposed method models priorities between items as a structured additive function (i.e., a function solely of an individual item's features and subject to identifiability constraints), rather than as a black-box policy dependent on the joint state of all items.

The reviewers appreciated the novel perspective introduced in this work and noted that the idea of leveraging additive decompositions appears to work well in practice. One reviewer also pointed out that the proposed method helps fill a real gap between index-based scheduling approaches and RL-based methods. Reviewers further highlighted that the experiments convincingly support the claim that effective transfer is possible across candidate sizes. Finally, two reviewers emphasized the importance of the theoretical analyses and guarantees introduced in this work.

At the same time, reviewers raised important concerns. Two reviewers questioned whether Assumptions A1 and A2 are realistic and likely to be satisfied in practice. During the discussion phase, the authors clarified some of the implications of each assumption and explained more clearly which parts of the method and analyses rely on them. Several reviewers also pointed out that many of the paper's claims were not properly supported in the submitted version due to the lack of comparisons with representative baselines. As submitted, the paper compared only against DQN, which is a single baseline and one that is not representative of the current state of the art in reinforcement learning. During the discussion phase, the authors conducted and reported additional experiments comparing against the Whittle index as well as other baselines such as NAM, A2C, PPO, TRPO, and QR-DQN, which the reviewers appreciated. Finally, one reviewer noted that, unlike what is implied in the paper, the proposed method appears applicable only to a limited class of problems. The authors responded that their approach was designed as "*a structured and transferable solution for a well-defined and practically important subclass of applications*". Nonetheless, the reviewer felt that this did not sufficiently address the concern, particularly because the claim made in the submitted manuscript was broader and appeared to suggest applicability to more general problems and MDPs.

Overall, all reviewers agreed that this paper addresses an important problem and explores a promising direction by leveraging ideas such as structured additive functions. They recognized that thoroughly evaluating this type of approach, which may have important real-world applications, is essential for supporting the paper's main claims, and they appreciated the various additional experimental comparisons conducted by the authors in response to the reviews. They encouraged the authors to further update the manuscript to address the limitations identified in the reviews, incorporate the new experimental results introduced during the discussion phase, and clarify possible sources of confusion, for instance regarding the implications of the work's main theoretical assumptions. All reviewers agreed that addressing these points would substantially strengthen the paper and help highlight its important contributions.